# Speed-up, slowdown, and redirection of ice flow on neighbouring ice streams in the Pope, Smith and Kohler region of West Antarctica

Heather L. Selley[1], Anna E. Hogg[1], Benjamin J. Davison[1], Pierre Dutrieux[2], Thomas Slater[3].

[1]School of Earth and Environment, University of Leeds, UK.

[2]British Antarctic Survey, UK.

[3]Department of Geography and Environmental Sciences, Centre for Polar Observation and Modelling, Northumbria University, Newcastle-Upon-Tyne NE1 8ST, UK.

*Correspondence to*: Heather L. Selley (H.L.Selley@leeds.ac.uk)

**Abstract.**

The ice streams feeding Dotson and Crosson Ice Shelves are some of the fastest changing in West Antarctica. We use satellite observations to measure the change in ice speed and flow direction on eight ice streams in the Pope, Smith and Kohler region of West Antarctica from 2005 to 2022. Seven ice streams have sped up at the grounding line, with the largest increase in ice speed at Smith West Glacier (87 %) whilst Kohler West Glacier has slowed by 10 %. We observe progressive redirection of ice flowlines from Kohler West into the more rapidly thinning and accelerating Kohler East Glacier, resulting in the deceleration of Kohler West Glacier and eastward migration of the ice divide between Dotson and Crosson Ice Shelves. These observations reveal previously undocumented impacts of spatially varying ice speed and thickness changes on flow direction and ice flux into downstream ice shelves, which may influence ice shelf and ice sheet mass change during the 21st century.

## 1 Introduction

Ice loss from Antarctica is dominated by increased grounding line discharge from ice streams draining into the Amundsen Sea Embayment (ASE) from the West Antarctic Ice Sheet (WAIS) (Joughin et al., 2010; Rignot et al., 2008, 2019; Shepherd et al., 2012, 2018). Combined, the ice streams in the ASE – Pine Island, Thwaites, Haynes, Pope, Smith, and Kohler Glaciers - contain 1.2 m of sea level rise potential (Morlighem et al., 2011; Mouginot et al., 2014), and are theoretically susceptible to Marine Ice Sheet Instability (MISI) due to being grounded on predominantly retrograde bed topography (Favier et al., 2014; Joughin et al., 2014; Schoof, 2007; Weertman, 1974). MISI is theorised to occur on marine ice sheets and is an instability where the ice is grounded below sea level on bed rock that slopes downwards into the interior of the ice sheet. This configuration has the potential to cause rapid retreat of the grounding line and increase ice flow into the ocean (Schoof, 2007). Across the ASE, satellite observations have shown widespread grounding line retreat (Konrad et al., 2018; Milillo et al., 2022; Scheuchl et al., 2016), thinning (Konrad et al., 2017; Pritchard et al., 2012; Rignot et al., 2019; Shepherd et al., 2018) and acceleration (Mouginot et al., 2014) since at least the early 1970's. It is likely these ice dynamic changes initiated in or before the 1940's, particularly on Pine Island Glacier (Clark et al., 2024; Davies et al., 2017; Rignot et al., 2014; Shepherd et al.,

2019; Smith et al., 2016). While ice loss from Pine Island and Thwaites Glaciers dominate the overall mass balance of the ASE, observations show that 21$^{st}$ century rates of thinning and acceleration at Pope, Smith, and Kohler (PSK) Ice Streams

have so far been proportionally faster and larger than that of their neighbours (Davison et al., 2023; Konrad et al., 2017; McMillan et al., 2014; Mouginot et al., 2014; Rignot et al., 2014; Surawy-Stepney et al., 2023b). Ice speed and grounding line discharge at Pope, Smith, and Kohler East Glaciers (Fig. 1) more than doubled between the 1970s and 2010 (Mouginot et al., 2014; Rignot et al., 2016). Smith Glacier thinned by up to 9 m/yr between 2010 and 2013 (McMillan et al., 2014), and its grounding line has retreated at speeds of up to 2 km/yr (Konrad et al., 2018) - the fastest rate recorded in Antarctica during the

satellite era.

The PSK region contains 8 distinct ice streams, two of which have dual branches extending ~100 km inland (Fig. 1). Haynes, Vane, Pope, Smith East, Smith West, and Kohler East Glaciers flow into the Crosson Ice Shelf, whereas Kohler West and Horrall Glaciers flow into the Dotson Ice Shelf (Fig. 1). Parts of Crosson Ice Shelf have doubled in speed, comparable to the

grounded upstream areas that now flow at speeds of ~1.2 km/yr, and the ice shelf has extensive rifting, particularly on its eastern side (Lilien et al., 2018; Mouginot et al., 2014) (Fig. 1). Both Dotson and Crosson Ice Shelves have thinned (Pritchard et al., 2012) by 10 % and 18 % (Paolo et al., 2015) between 1994 and 2012, with average basal melt rates of 5.4 ± 1.6 and 7.8 ± 1.8 m/yr between 1994 and 2018 (Adusumilli et al., 2020), respectively. Kohler West Glacier, which flows into the Dotson Ice Shelf, has changed less over the last 30 years, attributed to its prograde bed slope (Milillo et al., 2022; Scheuchl et al.,

2016). Satellite observations show that the grounding line of Kohler West Glacier has retreated 200 m/yr between 1992 to 2011 (Milillo et al., 2022), with no change in ice speed observed up to 2015 (Scheuchl et al., 2016) despite the ice shelf thinning near the grounding line (Gourmelen et al., 2017; Zinck et al., 2023). Despite the importance of the PSK region, due to its contemporary and potential contribution to sea level rise, the spatial pattern of its speed change since 2015 is not well characterised, and the consequences for coupled ice shelf-ice sheet dynamics are not adequately understood.


In this study, we combine new high-resolution satellite observations with existing measurements of ice velocity from PSK for 17.5 years from 2005-2022, to extend the record of ice speed and investigate the impact of spatially varying acceleration and thinning on ice flow direction and ice flux into Crosson and Dotson ice shelves. We measure the calving front location on Dotson and Crosson Ice Shelves from 2005 to 2020 to examine the impact of ice sheet velocity change on these two important

Antarctic ice shelves.

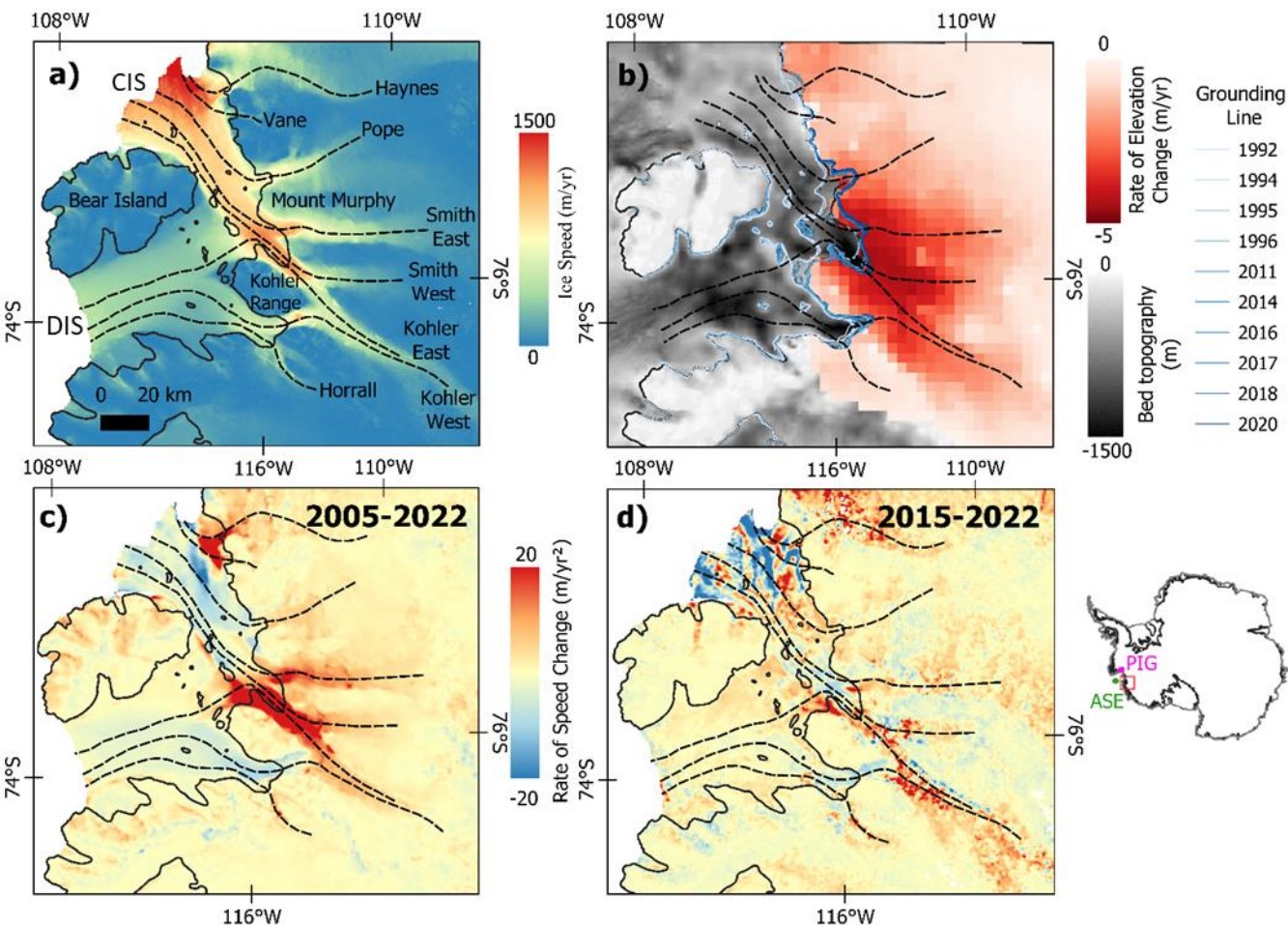

**Figure 1: Ice speed and rate of speed change at Haynes, Vane, Pope, Smith East, Smith West, Kohler East, Kohler West and Horrall Glaciers, which feed the Crosson Ice Shelf (CIS) and Dotson Ice Shelf (DIS) in the Amundsen Sea Sector of West Antarctica. (A)** Sentinel-1 SAR derived average speed 2015-2022. The 2011 grounding line location (solid black line) (Rignot et al., 2016) and the location of the 8 flow line profiles (dashed black lines) are also shown. **(B)** Rate of elevation change for 1992 to 2023 (red) (Shepherd et al., 2019) and grounding line locations from 1992 to 2020 (blue) (Milillo et al., 2022; Rignot et al., 2016). **(C)** Observed rate of speed change over the full 17.5-year study period, from 2005/05/01 to 2022/05/30. **(D)** Observed rate of speed change during the Sentinel-1 period from 2015/06/01 to 2022/05/31. Measurements are superimposed on BedMachine bedrock topography (Morlighem et al., 2017).

## 2 Method

### 2.1 Ice Velocity Measurements

We measured the ice speed of Haynes, Vane, Pope, Smith, Kohler, and Horrall Glaciers from January 2015 to December 2022 using Sentinel-1a and -1b synthetic aperture radar (SAR) satellite imagery (Table S2). Ice velocity was measured using the intensity feature tracking technique, which measures the displacement of visible features at or near to the ice surface such as crevasses or rifts and stable amplitude variations (Selley et al., 2021; Strozzi et al., 2002; Surawy-Stepney et al., 2023a; Wallis et al., 2023). Feature tracking was performed using the GAMMA software, with a window and step size of three window sizes of 256x64, 362x144 and 400x160 range and azimuth pixels, each with a step size of approximately 25 % of the window size, after image co-registration using precise (5 cm error) orbit ephemerides and a 1x1 km Digital Elevation Model (DEM) (Bamber et al., 2009). Errors in the velocity measurements are caused by errors in the auxiliary DEM, imprecise co-registration of the SAR images, uncertainty in locating the cross-correlation peak and atmospheric delay due to variations in tropospheric water vapour and signal propagation speeds in the ionosphere (Hogg et al., 2017; Nagler et al., 2015). We produce a spatially varying error estimate for each image-pair by dividing the ice speed by the signal-to-noise ratio of the cross-correlation function for each pixel (Lemos et al., 2018). The largest errors (> 30 %) occur at the rapidly deforming and highly crevassed shear margins and near each ice front, however, the centre of the major ice streams typically have a lower relative error (Fig. S2).

For each image pair, we generate a signal-to-noise ratio-weighted mean velocity field of all available cross-correlation window sizes after removing outliers in the 2-D velocity fields. To remove outliers in the velocity field generated using each window size, first we compare each speed field to the nearest MeASUREs reference speed map (Mouginot et al., 2019); speed estimates more than four times greater or four times smaller than the reference map are considered outliers and removed. Secondly, we removed velocity estimates with a signal-to-noise ratio (SNR) of less than 5.8 (De Lange et al., 2007). Third, flow directions more than 45 degrees different from the reference map are considered outliers and removed. Finally, we use a hybrid median filter with a 3x3 moving window, which replaces central pixels with the median of the central pixel, horizontally and vertically connected pixels, and diagonally connected pixels. Hybrid median filtering is designed to remove salt and pepper noise whilst preserving edges more effectively than a traditional kernel-based median filter. We generate 100x100 m annual ice speed and ice speed error maps, from 2015 to 2022, by mosaicking and averaging the image-pair speed and speed error estimates in each calendar year, in which each image pair is weighted by its SNR. Prior to annually averaging, we remove temporal outliers from the time series in two stages. First, we remove pixels more than 5 scaled Median Absolute Deviations (MAD) from the time-series median; we then repeat this filter with a threshold of 2. Secondly, we remove local time-series outliers, defined as speed estimates more than 3 scaled MADs from the local median in a 7-pixel moving window (approximately 6 weeks). After forming each annual mosaic, we further remove spatial outliers in two stages. First, we remove spatially isolated pixel groups, defined as contiguous regions of fewer than 100 pixels. Secondly, we remove pixels with unrealistically rapid changes in flow direction and speed, defined as pixels in which the flow direction differs by more than 35 degrees, or the speed differs by more

than 2 standard deviations, from the median of the surrounding pixels in a 5x5 kernel. Finally, we perform a second filter based on 8-directional speed gradients, where small (<1 km$^2$) regions enclosed by large speed gradients (more than 50 m yr$^{-1}$ over 100 m (1-pixel)) are removed.

100

We used the MEaSUREs 1x1 km Antarctic annual ice velocity data (Mouginot et al., 2017) to calculate the longer-term rate of ice speed change from 2005. This dataset was generated from a combination of SAR and optical imagery acquired over the period 01/06/2005 to 31/05/2017, using both intensity tracking and interferometric techniques (Mouginot et al., 2017) (Table S2 & S3). Combined with our new speed measurements, this study presents a 17.5-year long record of ice speed measurements from June 2005 to December 2022 (Fig. S2). For time series analysis, we extract ice speed measurements along flowline transects located on the fast-flowing central trunk of all 8 ice streams (Fig. 1a and Fig. 2) and compute grounding zone time-series averaging within 2.5 km diameter circles where the flow lines intersect with the grounding line (Rignot et al., 2016). To explore the shorter-term (sub-decadal) variability data were detrended by subtracting the linear trend. To establish how the change in flow direction impacts the ice flux into the ice shelves the upstream flux was calculated by seeding flowlines from the calving front in 2005 and 2019, then the flux calculated at every 100 m along each flowline. It was then integrated along each flowline and gridded for plotting and differencing. Velocity vectors were rotated parallel to the flowline to capture the mass flux every 100 m. A static thickness (Fretwell et al., 2013) was used to explore purely the impact of ice speed and flow direction changes.

## 2.2 Change in Calving Front Location and Ice Shelf Rifts

We measured the annual calving front location of the Crosson and Dotson Ice Shelves from 2005 to 2022 using Moderate Resolution Imaging Spectroradiometer (MODIS) imagery (Andreasen et al., 2023). Images were acquired from mid-January to the end of February to ensure consistent temporal sampling, to use relatively cloud free data, and to avoid aliasing seasonal variation. The calving fronts were manually delineated (Cook et al., 2005; Cook and Vaughan, 2010), and the annual resolution allows an overview of the calving front changes on the 17.5-year timescale. The uncertainty of the calving front location measurement is limited by the accuracy with which the boundary can be delineated and the accuracy of the image georeferencing. Annual ice shelf area measurements were calculated by using the annual calving fronts to modify a reference ice shelf mask (Rignot et al., 2011). To quantify any changes in the Crosson Ice Shelf, we measured the area where the calving front location extends inland of the compressive arch location from Lilien et al. (2018) assuming that the arch position is static throughout our study period. We also manually delineated a persistent, long (~12 km) ice shelf rift near Bear Peninsula using the Sentinel-1 amplitude images and calculated the distance between the rift tip's end and Bear Island's easternmost point.

# 3 Results

## 3.1 Ice Velocity

Our ice speed measurements show that, of the 8 major ice streams in the PSK region, 6 ice streams reached mean speeds of over 700 m/yr in 2022, with the fast-flowing regions of the ice streams extending up to 75km inland of the grounding line (Fig. 1a). The fastest flowing ice streams are in the centre of the PSK region, where Kohler East, Smith West, and Smith East Glaciers flow at speeds up to 1,215 ± 275 m/yr in 2022 (Table 1, Fig. 2). Relatively slower-moving ice is transported to the west of the inland Kohler Range through Kohler West and Horrall Glaciers, which flow at speeds of 715 ± 319 m/yr and 401 ± 173 m/yr at the grounding line in 2022, respectively. Ice is discharged through Pope, Vane, and Haynes Glaciers which flow at speeds of 772 ± 211 m/yr, 203± 93 m/yr and 810 ± 169 m/yr in 2022, respectively (Table 1).

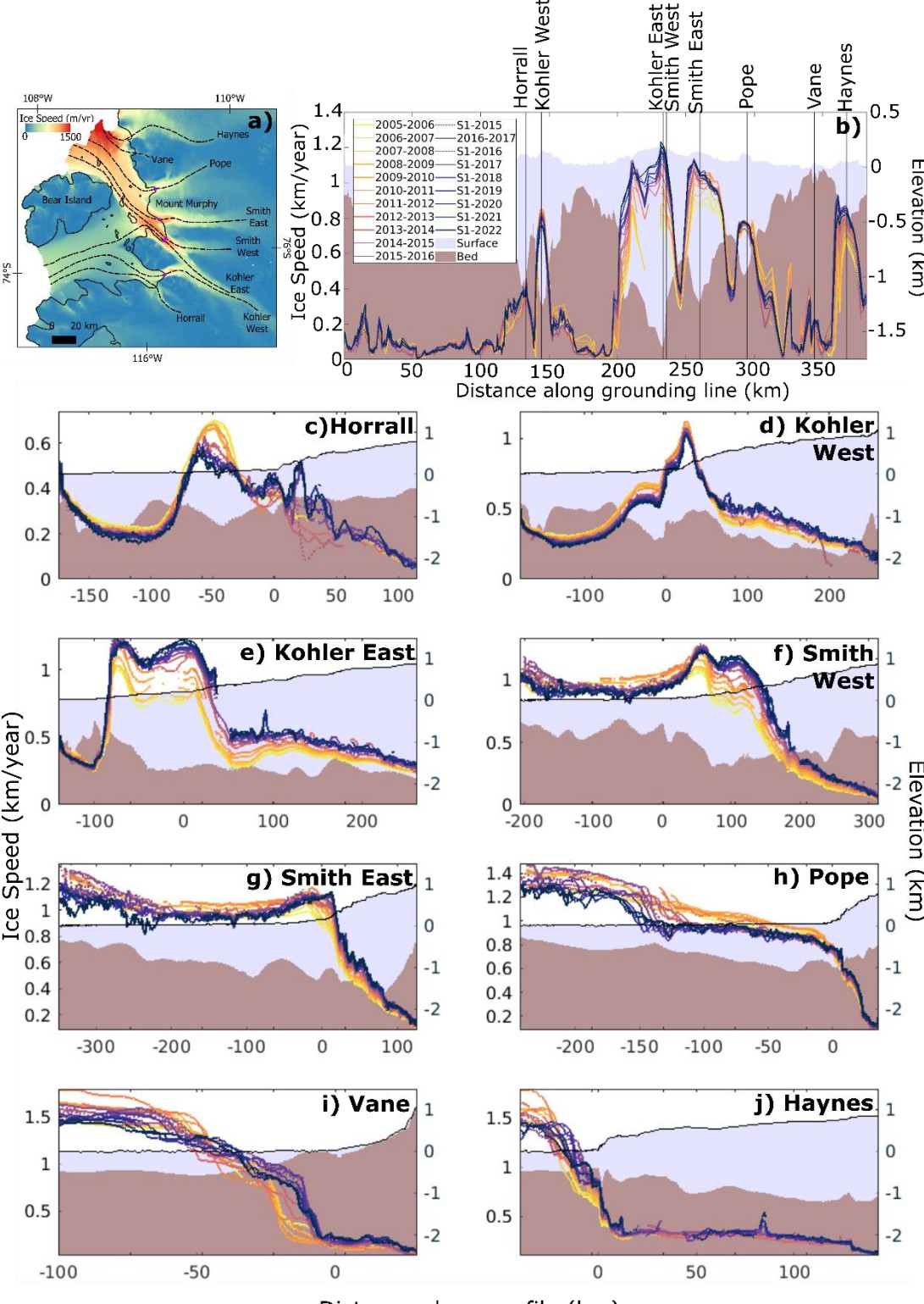

**Figure 2: (A)** Sentinel-1 SAR derived average 2015 to 2022 ice speed over the PSK region. The 2011 grounding line location (solid black line) (Rignot et al., 2016), the location of the 8 flow line profiles (dashed black lines) and the intersection 2.5 km buffer (purple circles) are also shown. **(B)** Observed ice speed from 2005 to 2022 at the 2011 grounding line (Rignot et al., 2016) of Dotson and Crosson Ice Shelves in the Amundsen Sea Sector of West Antarctica. The x-axis is shown as distance from the grounding line, with positive values indicating the inland section of the profile on the ice sheet, and negative values indicating seaward locations. Also shown are the intersections of flow lines (black vertical lines), the bed topography (brown shading) and ice surface elevation and thickness (grey shading) from BedMachine (Morlighem et al., 2017). Observed speed along the flowlines **(C)** Horrall **(D)** Kohler West, **(E)** Kohler East, **(F)** Smith West, **(G)** Smith East, **(H)** Pope, **(I)** Vane and **(J)** Haynes.

**Table 1:** The observed mean ice speed in 2022, rate of speed change 2005-2022 and during the Sentinel-1 period (2015-2022) and the percentage speedup for both periods and rate of surface elevation change (1992-2023) (Shepherd et al., 2019) for the 8 major ice streams in the PSK region.

| Region | Ice Stream Name | Ice Shelf the Ice Stream Primarily Feeds | Mean Ice Speed in 2022 (m/yr) | Rate of speed change 2005-2022 (m/yr²) | Rate of speed change for Sentinel-1 period (2015-2022) (m/yr²) | Speed change from 2005-2022 (%) | Speed change in Sentinel-1 period (2015-2022) (%) | Rate of Surface Elevation Change (m/yr) |
|---|---|---|---|---|---|---|---|---|
| Western | Horrall | Dotson | 401 ± 173 | 2 | 4 | 10 | 9 | -0.8 |
| | Kohler West | Dotson | 715 ± 319 | -5 | -2 | -10 | -3 | -1.1 |
| Central | Kohler East | Dotson | 1215 ± 275 | 32 | 7 | 84 | 5 | -2.5 |
| | Smith West | Crosson | 1188 ± 253 | 32 | 11 | 87 | 8 | -2.6 |
| | Smith East | Crosson | 1093 ± 416 | 14 | 0 | 30 | 0 | -2.7 |
| Eastern | Pope | Crosson | 772 ± 210 | 3 | 1 | 7 | 1 | -2.2 |
| | Vane | Crosson | 203 ± 93 | 5 | 5 | 76 | 22 | -0.9 |
| | Haynes | Crosson | 810 ± 169 | 17 | 10 | 60 | 11 | -1.1 |

## 3.2 Change in Ice Speed

We used our ice velocity measurements to fit a linear trend in each pixel to calculate the rate of change in ice speed from all speed data (MeASUREs and Sentinel-1) 2005 to 2022 and for the Sentinel-1 period 2015-2022 across the PSK region (Fig. 1b and 1c). We exclude pixels with fewer than 5 annual velocity estimates and retain speed change estimates where the fit was significant ($p<0.05$) (Selley et al., 2021). Our results show that 7 ice streams in the study region have sped up since 2005, with 4 of those ice streams speeding up by over 60 % at the grounding line (Table 1). We observe the largest percentage increase in ice speed at the grounding zone of Smith West Glacier, where the rate of ice flow increased by 87 % over the 17.5-year study period reaching speeds of 1.2 km/yr by 2022 (Fig. 3a and Table 1). The fastest absolute rate of speed change was observed on Kohler East and Smith West Glacier where ice speeds increased by 32 m/yr$^2$ from 2005 to 2022 (Table 1). High rates of speed change are also observed on Haynes (17 m/yr$^2$) and Smith East (14 m/yr$^2$) Glaciers (Table 1); the central region containing Kohler East, Smith West, and Smith East Glaciers, accounts for 74 % of all the observed speed-up at the grounding line (Fig. 1).

Our results show that there is significant spatial variability in the observed speed change signal across the PSK region, and that change on individual ice streams has not been linear throughout the 17.5-year study period. In the Central and Eastern regions of PSK, although ice flow has sped-up throughout the study period, the rate of speed-up since 2015 was lower than that observed in the preceding decade between 2005 and 2015 (Table 1). In contrast, ice flow on Kohler West Glacier was 10% slower in 2022 relative to its 2005 ice speed, with the slowdown starting around 2012. The western side of the Dotson Ice Shelf into which these ice streams flow into has also decelerated by -5 m/yr² since 2005 (Fig. 1a and b). On Kohler West Glacier, the rate of ice flow slow-down has increased since 2015 (- 8 m/yr$^2$) compared to the 2005 to 2015 mean (-4 m/yr$^2$) (Table 1).

In addition to these long-term trends, we find all ice streams in the PSK region exhibit clear short-term (sub-decadal) speed variability in addition to the 17.5-year trend (Fig. 3). This short-term variability is characterised by a period of rapid speed-up on all ice streams, excluding Kohler West Glacier, from 2005 to 2010 (14%) and from 2014 to 2017 (12%), albeit minimal for Horrall Glacier. These speed-up episodes were separated by periods of steady or decelerating ice flow from 2011 to 2013 (- 4%) and from 2017 to the end of the study period in 2022 (-2%) (Fig. 3a). There is some variability in the timing of these speed variations across the PSK region: slow-down begins later (2013) at Vane Glacier and is more prolonged at Pope Glacier (2012-2015), whilst the overall slow-down of Kohler West Glacier also began around 2012 but was sustained throughout the remainder of the study period.

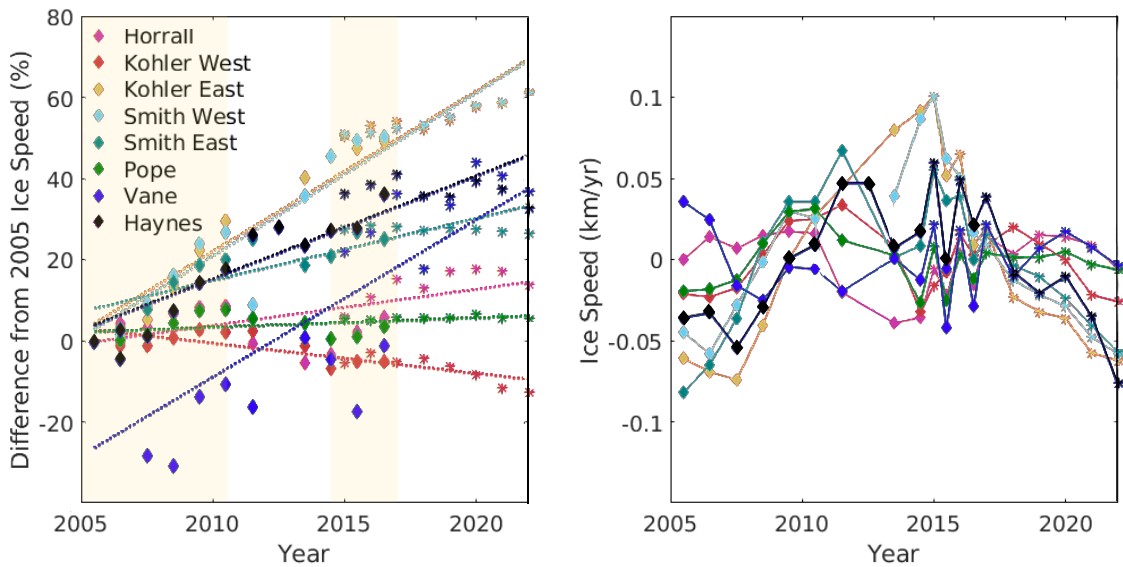

**Figure 3: Observed relative speed change on 8 major ice streams in PSK over the 15-year study period. (A)** Percentage change in speed from 2005 to 2022, from MEaSUREs (diamonds) (Mouginot et al., 2014) and Sentinel-1 (asterisks) velocity data, calculated as the difference in speed from the 2005 measurement. All measurements were extracted from a 2.5-km-diameter region where the central flow line crosses the grounding line (Fig. 1a) (Rignot et al., 2016). The shaded regions represent the rapid periods of speed up demonstrating the periodicity around the 17.5-year trend. **(B)** De-trended annual ice speeds for each of the 8 glaciers from 2005 to 2022, from MEaSUREs (diamonds) and Sentinel-1 (asterisks) velocity data, calculated as the difference in speed from the 2005 measurement. All measurements were extracted from a 2.5-km-diameter region where the central flow lines (Fig. 1a) cross the grounding line (Rignot et al., 2016).

### 3.3 Redirection of Ice Flow

Our ice velocity measurements reveal a substantial change in the direction of ice flow on two ice streams in the PSK study region (Fig. 4, S1 and Table S1), with three key changes in flow structure being especially notable. Firstly, at the division
170  between Kohler West Glacier and Kohler East Glacier, we observe an eastward rotation (~8˚) of the ice flow vectors by 2019 (Fig. 4d). Examination of ice flowlines generated from velocity measurements further show that this rotation results in redirection of ice flow from Kohler West Glacier into Kohler East Glacier, which we interpret as a form of 'ice piracy' resulting from the differential rates of surface lowering observed at this ice stream division (Fig. 4). Secondly, we observe a westward rotation of the flow vectors (~2˚) in the region just downstream of the grounding lines of Kohler East and Smith West Glaciers,
175  indicative of increased routing of ice into Dotson Ice Shelf that had previously fed the Crosson Ice Shelf (Fig. 4). Thirdly, the ice divide between Dotson and Crosson Ice Shelves has migrated eastwards within the study period (Fig. 4). These changes in

flow direction have had an impact on the advection of ice from Kohler West and Kohler East into Dotson and Crosson Ice Shelves, resulting in increased flux into Dotson and a smaller increase in flux to Crosson than would have been the case had flow directions remained stable (Fig. 4).

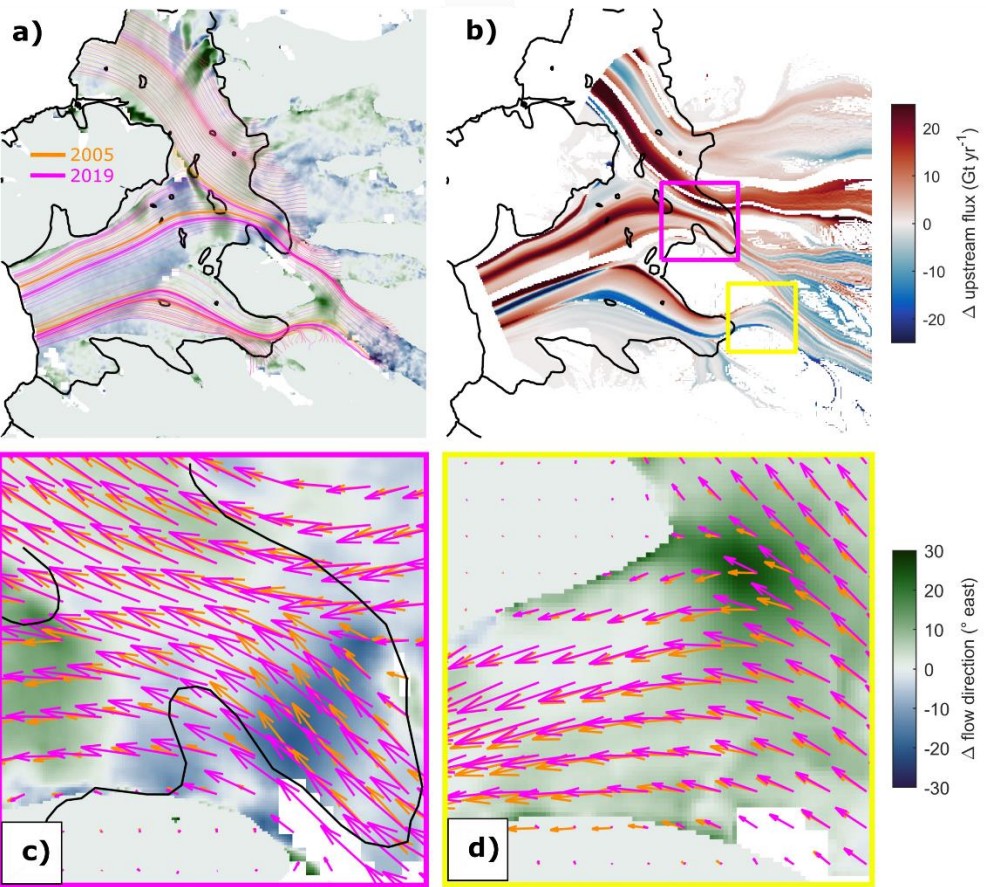

**Figure 4: Flow lines and direction changes in the central Pope, Smith and Kohler Region. (A)** Flow lines at the grounding line of the central PSK region for 2005 (orange) and 2019 (pink). Flow direction change (colour shading), and the grounding line location (solid black line) (Rignot et al., 2016) are also shown. **(B)** Change in upstream flux 2019 compared to 2005 calculated and displayed cumulatively along the flowlines from inland to the calving front, which indicates where ice mass is predominantly being directed on the ice shelves. Calculated using a time-varying ice thickness drawing on elevation change observations (Shepherd et al., 2019). Panels **(C)** and **(D)** show flow direction change and flow vectors in sub-regions near **(C)** the grounding line of Kohler East Glacier and **(D)** the division between Kohler West and Kohler East.

**3.4 Change in Calving Front Location and Ice Shelf Rift Growth**

Our measurements of the Dotson Ice Shelf calving front location show that the ice edge has remained in a relatively constant position throughout the study period, with a small (~4 km) but constant advance of its eastern limit until 2021 and a slight (~5 km) retreat in the west from 2005 to 2022 (Fig. 5). In contrast, the calving front of Crosson Ice Shelf has undergone several phases of retreat since 2005, the largest of which occurred in the period between 2013 and 2014. Overall, the calving front retreated by ~60 km, losing a ~85 km$^2$ area inland of the 1996 compressive arch (Lilien et al., 2018) at the eastern shelf edge and becoming heavily crevassed. During the same period as these large calving events, a 5 km rift formed at the eastern margin of the Crosson Ice Shelf and grew to ~12 km across the width of the ice shelf towards Bear Island during the study period (Fig. 5b, S1and S3). The distance between the end of the rift and Bear Island decreased from 8 km in 2015 to 5 km in 2022, with the rate of rift growth slowing from 2016 to 2022. If the rift eventually reaches Bear Island a ~4 km$^2$ iceberg may be released. After the 2014 calving event, the shape of the shelf remains convex and in a fairly consistent configuration, with small further inland penetration, for example in 2016 (~10km further). This positioning is maintained with additional fracturing and rifting gradually penetrating further across to the western side of the shelf. Notably, in 2021 there is an advance in the floating ice in front of Haynes Glacier which partially recloses the open pathway of water accessing the eastern side of the Crosson Ice Shelf. In 2022, this configuration persists however the shear margin in front of Mount Murphy fractures and partially detaches. The substantial visible increase in damage on the Crosson Ice Shelf inland of the compressive arch will have reduced the buttressing strength provided by the ice shelf, contributing to the observed speed-up of the glaciers that drain into it (Fig. 5).

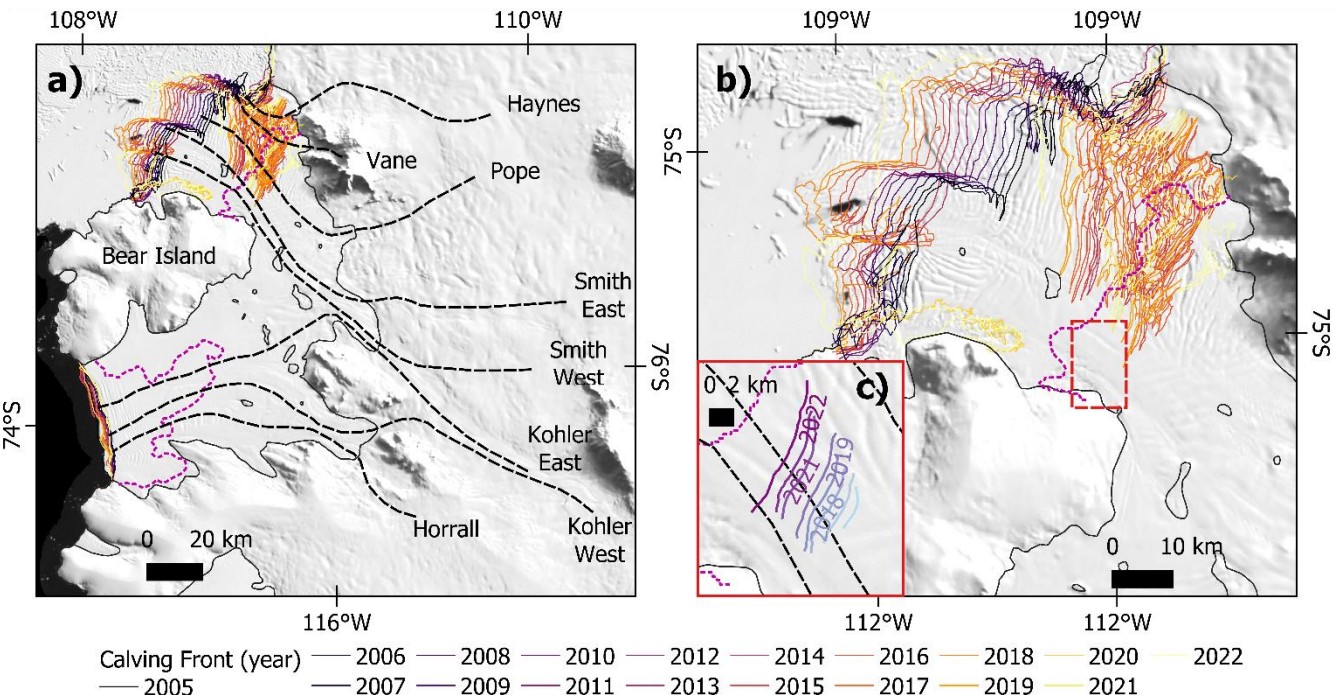

**Figure 5: Dotson and Crosson Ice Shelf calving front and rift location.** (A) Dotson and Crosson Ice Shelf calving front locations from 2005 to 2022 and the compressive arches (dotted pink line) (Lilien et al., 2018), are shown superimposed on the MODIS mosaic (Scambos et al., 2007). (B) Zoom of Crosson Ice Shelf calving front locations from 2005-2022. (C) Location of the rift approaching bear island through the Sentinel-1 period (2015-2022).

## 4 Discussion

Overall, our results extend the record and show a long-term speed-up at most major ice streams in the PSK study region from 2005 to 2022, with an average speed increase at the grounding line of 51 %. Previous work focused on the longer-term changes back to the 1970s (Mouginot et al., 2014); here we focus on the short-term changes visible from the annual records available from 2005. Although speed-up is greatest at the grounding line (Mouginot et al., 2014) it also extends up to ~100 km inland (Fig. 1b, 1c), impacting a substantial portion of the West Antarctic Ice Sheet. The zone of highest relative speed-up is located inland of Bear Island on Smith West Glacier where the ice is thickest (Lilien et al., 2018), the bedrock is deepest (Schoof, 2007) and where the grounding line has retreated the most (Konrad et al., 2017) (Table 1, Fig. 1b, 1c and Fig. 2). During their overlapping periods, our observations of speed change closely agree with previous studies in terms of the magnitude and spatial patterns of speed change (Lilien et al., 2018; Miles et al., 2022; Mouginot et al., 2014; Scheuchl et al., 2016), which are consistent with observed patterns of grounding line retreat (Rignot et al., 2014) and surface elevation change (McMillan et al., 2014). Our examination of individual ice stream behaviour and our addition of more recent observations add further detail to our understanding of the evolution of the PSK region.

## 4.1 Short term periodicity

In addition to the long-term speed up, we observe shorter-term ice speed variability superimposed on the long-term speed-up of most glaciers in the study region. We note periods of rapid speed-up 2005 to 2011 with an average speed difference across all ice streams of 14 % and during 2014-2017 (12 %) were interrupted by periods of slow down between 2011-2013 (-4 %) and 2017-2020 (-2 %) (Fig. 3a); the timing and relative magnitude of these speed fluctuations vary between ice streams, but the absolute magnitude of the speed change is similar across the region (Fig. 1). Similar ice speed variability during these periods has previously been documented at Pope Glacier, Kohler West Glacier, and Pine Island Glacier (Lilien et al., 2018; Mouginot et al., 2014; Scheuchl et al., 2016). At Pine Island Glacier, this ice speed variability is attributed to fluctuations in the depth of the thermocline on the continental shelf (Assmann et al., 2013; Dotto et al., 2019; Walker et al., 2007; Webber et al., 2017). Until 2014, the shorter-term speed variability at PSK was broadly synchronous with those observed at Pine Island Glacier, implying that the thermocline depth changes and the associated changes in oceanic heat delivery to ice shelf grounding lines during those times were widespread in the Amundsen Sea Embayment, consistent with hydrographic surveys (Jenkins et al., 2018) and numerical ocean modelling (Naughten et al., 2022a). Pine Island Glacier has rapidly sped up since 2017 (Davison et al., 2023; Joughin et al., 2021), whereas in PSK rates of speedup have slowed or ice speeds have plateaued (Fig. 3). This may reflect the increased role of calving and damage (Lhermitte et al., 2020; Surawy-Stepney et al., 2023b) on the recent speed changes observed at Pine Island Glacier (Joughin et al., 2021). Particularly, the rate of speedup decreasing after 2020 may also be linked to the advance in the floating ice in front of Haynes Glacier which partially recloses the pathway of open water accessing the eastern side of the Crosson Ice Shelf perhaps limiting the penetration of the warmer water into the cavity. A re-buttressing of the Crosson Ice Shelf may have also occurred, albeit in a new configuration, with the calving front connecting to Mount Murphy as a pinning point and lowering the rate of observed speed up. Another contributing factor could be the ungrounding and re-pining of ice rises beneath the ice shelves. Notably, there are two rises directly downstream from Kohler East and Smith West, and should their pinning strength change, this would result in a significant speed change. Change in ice shelf pinning can cause destabilisation through reducing the structural stability of the shelf when they are in advanced stages of thinning (Benn et al., 2022) but also by potentially initiating calving events (Arndt et al., 2018). These processes may impact the ice shelf at different times in different regions, potentially driving some of the observed speed change (Miles and Bingham, 2024). It also remains unclear whether there is a pathway for the ocean to flow between the Crosson and Dotson Ice Shelves, and how this pathway has evolved historically.

## 4.2 Kohler West slow down

In contrast to the rest of the region, Kohler West Glacier did not speed up appreciably after 2013, resulting in a moderate (~10 %) slowdown since 2012. The lack of significant speed-up of Kohler West Glacier has been attributed to the prograde bed slope on which the grounding line rests (Milillo et al., 2022), making the ice stream less susceptible to ice dynamic change and instability. The region continues to thin and therefore thinning induced reductions in driving stress may have contributed

to the observed slow down. Indeed, 18 m of thinning and a 0.03-degree decrease in ice surface slope (comparable to those observed on Kohler West) would cause an 82 m/yr decrease in ice velocity, which is similar to the observed slow-down from 2005 to 2022 (Equation S1, Cuffey and Paterson, 2010). In conjunction with this deceleration of Kohler West, we observe redirection of ice flow from Kohler West Glacier into Kohler East Glacier (Fig. 4). We suggest that this so-called 'ice piracy' results from the differential thinning rates of Kohler West (less than 2 m/yr) and Kohler East Glaciers (up to 9 m/yr) over the last few decades (McMillan et al., 2014; Nilsson et al., 2022). These changes in flow direction are consistent with the observed changes in surface elevation and ice thickness in this region and would be expected to occur elsewhere in places where the geometry permits, such as at Totten and Vanderford Glaciers (McCormack et al., 2023). Analogous historical changes in flow direction and ice divide migration have been inferred from chemical tracers (Iizuka et al., 2010) and are widely documented in paleo ice sheet reconstructions (Catania et al., 2012; Conway et al., 2002; Conway and Rasmussen, 2009; Iizuka et al., 2010), whilst ongoing ice divide migration has been inferred from isochrones (Nereson et al., 1998) and thickness change measurements (Conway and Rasmussen, 2009). To our knowledge the redirection of ice flow from one ice stream to another has not been observed directly on ~15-year timescales, although, it may have occurred at glacier and ice cap scales. Differential thinning near ice stream boundaries is not isolated to this study region; therefore, we expect that ice divide migration and the associated changes in flow direction have and will occur elsewhere in Antarctica, and this should be considered when interpreting observed changes in ice speed and ice thickness in tightly connected regions. Modelling work indicates that minor changes in ice sheet geometry can result in 'ice piracy' and highlights the importance of quantifying the rates of change and subsequent impact on ice mass loss (McCormack et al., 2023).

### 4.3. Crosson and Dotson Ice Shelves

The observed changes in flow speed and flow direction have downstream consequences for Dotson and Crosson Ice Shelves. The slow-down of Kohler West Glacier has reduced the ice flux into Dotson Ice Shelf by 0.68 Gt (11 %) locally from 2011 to 2022 (Fig. 4b). However, the discharge from Kohler East Glacier into the eastern part of Dotson Ice Shelf increased by 5.6 Gt (26 %) over the same time period because of the acceleration of Kohler East Glacier and redirection of ice flow resulting in an eastward migration of the ice divide between Dotson and Crosson Ice Shelves. We suggest that the divide migration between Dotson and Crosson Ice Shelves has been instrumental in maintaining the apparent stability of the Dotson Ice Shelf, despite the slow-down of Kohler West Glacier. In contrast, Crosson Ice Shelf deteriorated substantially during our study period, characterised by rapid thinning and extensive crevassing, which we suggest has been exacerbated by the redirection of ice from Kohler East due to the ice shelf divide migration. There has been significant variability in sub-shelf melt rates in the Crosson Ice Shelf spatially and temporally over the past 20 years, with melt peaking in the early 2010s (Jenkins et al., 2018). Additionally changes to pinning points can further cause destabilisation through initiating calving events (Arndt et al., 2018) and in advanced stages of thinning (Benn et al., 2022). Further work exploring the chronology of changes to stress structures on the Crosson Ice Shelf would be invaluable to further establish the interplay of these driving mechanisms.

We do not observe ice piracy occurring in any of the other ice streams in the region (Figure 4). This is likely due to their geometry which does not result in the spatial gradient of thinning we observe at Kohler West/East junction.

**4.4. Potential implications**

Some previous research has suggested that the rapid grounding line retreat and acceleration in the PSK region is indicative, if not diagnostic, of MISI (Milillo et al., 2022). At Pine Island Glacier, the rapid reduction in ice velocity in response to the cold-water intrusion in 2012 to 2013 and presumably reduced basal melt rates has been used as evidence to suggest that MISI is not underway, or at least is conditional upon continued ocean forcing (Christianson et al., 2016). Recent modelling efforts suggest

that the current grounding line positions in the Amundsen Sea Embayment (ASE) are stable (Hill et al., 2023). The fluctuations in speed we observe suggest that ice streams in this region are responding to changes in external forcing similarly to Pine Island Glacier and that MISI may also not be a major dynamical factor at short (sub-decadal) timescales across the PSK region. Conversely, more recent modelling work based at Pine Island Glacier indicates irreversible grounding line retreat between the 1940s and the 1990s is an example of MISI with recent changes being primarily driven internally (Reed et al., 2024). We

observe that ice streams on prograde (Kohler West) and retrograde bed topography are responding with opposing speed change trends (Fig. 2). This indicates that bed geometry is a key control on processes driving speed changes. It is plausible that there is still an instability which could have been triggered on centennial scales. The speed variability is indeed superimposed on a longer-term speed-up. This may be driven by a centennial trend of ocean warming on the continental shelf (Naughten et al., 2022b), likely driven by a trend in winds over the continental shelf break (Holland et al., 2019, 2022), or it could be driven by

internal ice dynamic feedbacks (such as MISI)(Reed et al., 2024) or ice-ocean feedbacks, such as changes in cavity circulation due to ice shelf thinning and grounding line retreat (Bradley et al., 2022).

**5 Conclusions**

Our 17.5-year-long record of ice speed shows continued speed-up on the majority of ice streams in the Pope, Smith and Kohler catchment of West Antarctica, slow-down on two ice streams and previously unreported changes in flow direction and ice

divide locations. We measure concentrated regions of speed-up (51 % on average) at the grounding lines of Kohler East, Smith West and Smith East Glaciers since 2005. In contrast, Kohler West slowed down by 10 % over the 17.5-year period, with most of this occurring since 2012. This is potentially due to thinning-induced reductions in driving stress. Superimposed on these speed change trends is a notable and widespread shorter-term periodicity, which is likely driven by variations in the depth of the thermocline in the ocean water on the continental shelf, comparable to that observed in the vicinity of Pine Island Glacier.

In addition to these speed changes, we also observe substantial ice flow reorganisation of both grounded and floating ice. This has been characterised by 1) redirection of ice flow from Kohler West into Kohler East Glacier, which we attribute to differential thinning of the two ice streams, and 2) migration of the ice divide between Dotson and Crosson Ice Shelves, which has resulted in redirection of ice flow from Kohler East and Smith West Glaciers into Dotson Ice Shelf from Crosson. These changes in flow direction have substantially altered the mass flux into Dotson and Crosson Ice Shelves, likely playing an

important role in maintaining Dotson Ice Shelf and accelerating the deterioration of Crosson Ice Shelf. This suggests that ice flow redirection is an important component of contemporary ice sheet dynamics which is required to understand the recent structural change in Dotson and Crosson Ice Shelves, and which may affect their future evolution. Together, these observations reveal previously undocumented interactions between the floating and grounded ice which will affect the future sea level contribution from the PSK region.


**Acknowledgments**

This work was led by the School of Earth and Environment at the University of Leeds. The authors gratefully acknowledge the European Space Agency, the National Aeronautics and Space Administration, the Japan Aerospace Exploration Agency

and the Canadian Space Agency for the acquisition of ERS-1 and -2 (C1P9925), Sentinel-1, Landsat-8, ALOS PALSAR and RADARSAT data, respectively. We acknowledge the use of datasets produced through the NASA Measures programme for funding the development of long-term climate data records from satellite observations.

**Funding:**


European Space Agency via the ESA Polar+ Ice Shelves project grant ESA-IPL-POE-EF-cb-LE-2019-834 as part of the ESA Polar Science Cluster (AEH, BJD).

European Space Agency vit the SO-ICE project grant ESA AO/1-10461/20/I- NB as part of the ESA Polar Science Cluster (AEH, BJD).

Natural Environment Research Council via the DeCAdeS project NE/T012757/1 (AEH, BJD, PD)

UK Earth Observation Climate Information Service NE/X019071/1 (AEH, BJD).

NASA provided support through grant 80NSSC20K1158 (PD).

**Author contributions:**


Conceptualization: HLS, AEH

Methodology: HLS, BJD, TS

Investigation: HLS, BJD

Visualization: HLS, BJD

Supervision: AEH

Writing—original draft: HLS

Writing—review & editing: HLS, AEH, BJD, TS, PD

**Competing interests:** All other authors declare they have no competing interests.


**Data and materials availability:**

The annual ice speed mosaics, rate of ice speed change, calving fronts and Bear Island crack data will be made available from Pangaea upon publication.

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
