# Peer review of "Speed-up, slowdown, and redirection of ice flow on neighbouring ice streams in the Pope, Smith and Kohler region of West Antarctica"

_EGUsphere, 2024_

## Referee Comment (RC1)

Review of *Speed-up, slowdown, and redirection of ice flow on neighbouring ice streams in the pope, smith, and Kohler region of West Antarctica by Selley, Hogg, Davison, Dutrieux, and Slater*

*Overview*
This study characterises changes in ice flow in the Pope, Smith, and Kohler region of West Antarctica from 2005-2022 using satellite observations from Sentinel-1a and 1b combined with MEaSUREs estimates.

The manuscript is well written, aims clear, and the analysis well executed. I don't have any major issues, and I suspect the points that I've highlighted below should be relatively straightforward to address.

My main issue is that I came away a bit murky about what's driving what changes in this region and how that relates to instability (this could be partly because I'm less familiar with the history of this region!). For example, the Crosson Ice Shelf has had some very large ice front retreat associated with large calving events, the margin between Dotson and Crosson has migrated eastwards, and there has been an increase of flux into Crosson. But I found the discussions around the drivers of these changes (i.e. the sequence of changes and causation) and how they relate to the overall stability of the system a bit unclear. Stepping through these aspects in the discussion in a bit more detail for both Crosson and Dotson (i.e. in paragraphs starting on L233, L245 and L255) would be helpful.

Some more specific comments are as follows:
- L30: "While Ice loss" → "While ice loss"
- Section 2.1. Could you add a couple of sentences about the Sentinel data, including the overall timeframe of the data, resolution, and image pairs considered.
- Figure S3. It'd be great to get subtitles on each figure panel and to include a 3rd column that shows the % differences in flow speed. It's otherwise hard to eyeball a 30% error in the shear margins (ref L78 in manuscript)
- L81-100: Removal of data points. It would be helpful for a non-expert for a few extra details here:
    - What % of the data were removed (including outliers etc)? Out of interest, are these points somewhat randomly distributed over the whole study region or do they concentrate?
    - L85: Some words about why the values of 5.8 SNR and 45 degrees were chosen would be helpful
    - Is the approach that you've taken standard (asking as a non-expert)? It's not essential, but I wonder if a schematic of the workflow could be helpful to readers who aren't very familiar with the procedures
- Can you make the supplementary figure references chronologically ordered in the manuscript? Currently fig. S3 comes before S1 and S2 and table S2 before S1
- L101-105: is there good agreement between MEaSUREs velocities and those estimated here using Sentinel for the overlap time period (2015-2017)? A comment

on this, or perhaps a few sentences in the supplementary information, would be helpful.

- L117: Is Fig. 1e missing or is this meant to reference a different figure?
- Figure 2, panels c-j: It's difficult to orient these lines and to know where they start and where they finish. What does 0 km (x-axis) represent? It would be helpful to be able to compare with figure 3 from Milillo et al. (2022) re the discussion on the importance of the prograde slope of Kohler for the stability of this ice stream (ref L234-236). Also, are the ice speed tick labels on the left y-axes of panels c-j correct? The speeds are low.
- Figure 3: what are the shaded regions in figure 3a?
- L163: by what degree do the ice flow vectors rotate in the piracy from KW to KE? From figure 4 it looks like it's mainly restricted to <|30 degrees|, but there are some regions where the colour bar saturates in figure 4d. I'm thinking back to the removal of ice velocities that have > 45 degree rotation and wondering if data from some pixels could be removed inadvertently?
- Figure 4b: It would be helpful to see the change in thickness over the same period so that it's clear where flux changes are due to flow piracy vs dynamic thinning. There are conflicting labels: "downstream flux" in the caption, but "upstream flux" in the colour axis label.
- Figure 5a: re the colours of the calving front contours in the Crosson Ice Shelf – was there retreat and then readvance to near the original calving front position in the latter part of the time period? It's a little bit difficult to date the different positions with the colour map used, and I'm uncertain whether the calving front has continued to retreat over this period. This would be helpful to know and to be able to visualise as it could help with the interpretation of the post-2014 behaviour in figure 3. Perhaps a zoom in of the calving front position here could be helpful, and some indication of the timing of retreat / readvance with labels. Also, are the figures in vector format? I couldn't zoom in to see the details.
- L217-219: I'm not sure I understand the sentence, particularly the references to "interrupted" on L219.
- L216-231: It would be good to link the variability post-2014 of the ice streams that feed Dotson to the calving front position in this discussion, particularly if it's possible to attribute (at least qualitatively!) changes in the velocities to calving and ocean forcing separately. I would have imagined that the retreat of the calving front into the compressive arch region could have caused more marked speedup (I'm assuming that there has been some more ice front retreat after 2014), but that is not reflected in the trends (figure 3) where the PSK rates of speedup have decreased (generally, although that wasn't the case for Horrall). Why do you think this is the case? Or do the speed changes post-2014 link directly to the thermocline variability (here, a decrease in melt rates)?
- L249-254: What do you mean by the divide migration between Dotson and Crosson having maintained the stability of the Dotson Ice Shelf? Also, it'd be good to elaborate on how the redirection of ice from Kohler East has exacerbated Crosson deterioration.

---

## Referee Comment (RC2)

**Review of "Speed-up, slowdown, and redirection of ice flow on neighbouring ice streams in the Pope, Smith and Kohler region of West Antarctica"**

Selley et al. document a velocity change in 8 glaciers that feed the Dotson and Crosson ice shelves over a recent 17.5 year period. They found most of the glaciers accelerated, with the exception of one tributary, Kohler West, which slowed down. This resulted in a change in the location of the ice divide separating the two ice shelves. The authors infer one step further and suggest the change in the mass flux into the two ice shelves contributed to the disintegration of part of the Crosson ice shelf.

The Amundsen Sea Embayment is a "hot topic" for Cryospheric studies due to the warming of the region and the sea level rise potential of the outlet glaciers. This study contributes unique findings on the less studied glaciers of the region (relative to Pine Island Glacier and Thwaites). However, some issues need to be addressed prior to publication. Please see my comments below.

**Major comments**

Structure of the text:
I found that several areas of the manuscript text needed more clarification or slight reorganization.

Section 2.1 is highly detailed. It is refreshing to see the intricacies of the analysis; however, it is not necessary to include everything described in the main text. I suggest including the specifics in the supplemental information and keeping the main text clear and concise.

Section 3.3 the possibility of "ice piracy" and its relationship to other "ice piracy" events. This is a fascinating topic and I believe this would fit much better in the discussion. Most past papers on ice piracy have invoked changing subglacial hydrology, but I think using the term here is fine. Notes posted in the marked-up text suggest a few additional references.

The discussion requires more coherent organization and clearer key take-aways. There were many aspects of the results that felt thrown into the discussion (e.g., Lines 218-220 – this observation should be mentioned in Results first). I suggest reorganizing the paragraphs into three main sections: Long-term trends – where you compare to PIG and MISI; Short-term periodicity – where you discuss the variability in the velocity on the annual scale and potential causes; and the connection to Dotson/Crosson – the "ice piracy" event, movement of the ice divide, and the subsequent deterioration of Crosson. It would also be interesting to see more information in the discussion about the ice shelf ice front and the rift propagation, and any connection between those results and the velocity results.

A quick note on the short-term periodicity – could it be driven by ungrounding of the numerous ice rises in the Crosson-Dotson (especially the latter) system? It is important to entertain the other possibilities that it is not related directly to the ocean dynamics – without more oceanographic observational data in this local region and time period. It seems far more likely to me that the loss of the pinning points, indirectly related to warm ocean conditions at depth (and

not necessary linked to ocean circulation events at PIG) are the cause of the rapid speed-up periods.

Regarding the very interesting "ice piracy" event - it would be beneficial to include more information about the thinning that occurred specifically during the period of the ice flow change. Additionally, it would be advisable to discuss the widespread thinning throughout the region and why the diversion of flow did not happen there.

I also found the font to be blurry on most of the figures, I suggest making sure the plots are of a higher DPI (300 minimum), especially in the supplemental figures.

Figures:
The figures in the text show a lot of important information, but are a bit hard to follow in some cases. For all of the figures that show spatial data it would be better to move the legend scalebar into each respective panel -- following which scale bar belongs to which panel takes significant effort.

Figure 1: I suggest making each panel larger to fit more information on the panel itself. For example, each panel needs a label and legend. I suggest moving the profile line labels/glacier names so that they are more clearly associated with each profile line. I'm not sure the 'bed topography data is very useful – it does not show up under any ice-covered area (where it would be more interesting – either it should have its own panel or perhaps just remove it. If you keep it, re-consider the scale – perhaps just -3000 to 0 meters is adequate? The 0-25km scale should be located on one of the panels. I suggest including an Antarctica overview map in one of the inset panels (preferably with ASE and PIG labelled since you discuss these in detail). The period of the "rate of speed change" should be labelled on the panels

Figure 2:
I suggest labeling the profile in 2b with 'Grounding Line profile', and also increase the size of the font of the glacier names and center them on the panels. You can gain space for your graphic panels by putting the labelling letter inside the panel (e.g. bottom left corner). The ice speed legend could be inside panel A. Panels C-J could be expanded to be the same size as the upper two panels (see below – red arrow) and tighten the spacing between panels C-J. Make the "distance along gate" legend larger (now that the scale bars are moved), and change "gate" to profile. In legend, "surface" and "bed" could be on the panels themselves to make the figure space more efficiently used.

[Figure]

Figure 5 – The calving front is difficult to distinguish for Crosson, especially in the north west corner. I suggest a zoomed in inset. The scale bar legends should be placed into the panels and Panel B made larger. The rift time series could be labeled with the year instead of the color bar (It would be clearer this way and since there are only 7 years and it won't take up too much space – especially when you make the panel larger). Scale bar is too small – make a bit longer and less bold (0-50 km? with ticks instead of a solid black square). I suggest labeling Bear Island on both panels. Place panel label "B" in top right corner to match panel A. Specify panel B is the orange rectangle in panel A.

Figure S1 – Please see the comment about the DPI quality of the figure. I also suggesting aligning the legend scales with the respective panels. I am wondering how panels H-J are different than Figure 4?

**Minor comments**

There are more significant figures than can be justified - rounding to the nearest 10th for the error and velocity measurement would be better.

Line 26: I suggest including a definition of MISI.

Lines 30-36: I think it is better to introduce dynamic details after the introduction of geography of the area. See comment in general comments. Also, A newer paper by Clark et al., 2024

provides further support for the 1940s kick-off for ice loss
https://doi.org/10.1073/pnas.2211711120

Lines 41-42: It would be better to include this before line 36 before talking about Kohler dynamics. Throughout this paragraph I suggest to only use East and West *Glacier Name* instead of alternating between East/West/Singular Glacier Name. It is unclear which tributary of the glacier stream you are referring to when you only use the singular name.

Lines 42-43: I suggest adding rifting detail "ice shelf has extensive rifting, particularly on its eastern side" after line 43-45.

Line 45: I suggest revising this sentence "Kohler West Glacier and the Dotson Ice Shelf into which it flows have changed less over the last 30 years" to – "Kohler West Glacier, which flows into the Dotson ice Shelf, have changed less over the last… attributed to the prograde slope.."

Lines 45-50: I suggest revising these sentences as the structure is a bit confusing.

Line 51: I think you mean this '...speed change since 2015 is not well characterized, and....?

Line 55: for 17.5 years from 2005-2022.

Figure 1 caption – I suggest shortening the figure caption. For example, "Average 2015 to 2022 ice speed over the PSK region, measured using interferometric Wide (IW) mode synthetic aperture radar (SAR) data acquired by the Sentinel-1a/b satellites." It would be sufficient to say "Sentinel 1 SAR derived averaged ice speeds over PSK region from 2015-2022." It is also not mentioned where the velocities from 2005-2015 come from – I thought they were from 2005-2022? It alternates throughout the text, please be consistent. Additionally, please write the dates with backslashes or dashes instead of periods. The periods make it look like a number and not a date. Please fix this throughout the text. As a quick note, if you include the method of measurement for one panel in the caption please then do so for the others – however, I suggest not including it in the caption and reserve the details for the manuscript text.

Line 69: "which" should have a comma beforehand

Line 70: Here you list features and then say "stable amplitude variations", which is not a feature.

Lines 72-74: I suggest including this in the supplemental information.

Line 76: I suggest changing to "...and signal propagation speeds in the ionosphere… etc"

Lines 78-79: please report the lowest error and average error. Additionally, Figure S2 was not referenced before, so I suggest changing S3 to S2 and correcting the text and supplemental information to have the correct figure numbers.

Figure S3: Please clean up the figure and follow suggestions made in Figure 1 and general comments (e.g. legends, scale, etc. also, the lat/lon does not need to have the minutes or seconds included as they are all 0. Keep it simple and clear).

Line 83: I am wondering which reference speed map is this? How is it different than the MEASURES data also used? I suggest restructuring the sentence to include "First" so it matches the rest of the paragraph.

Line 84: I suggest removing passive voice (ie. rewrite as: "we remove..etc")

Line 104: See comment about dates.

Lines 110-112: Can you comment on the limitations of only using imagery from 1-2 months of the summer season?

Line 117: There is not a Figure 1E.

Line 121: You use both flow units and ice streams here. What delineates a "flow unit" if not an ice stream? Please use consistent terminology. Additionally, what is meant by ice streams penetrating 75 km inland of grounding line? What happens after that?

Line 125: Which part of the time series was used for the averaging? I also suggest putting these circles on Figure 1 or in a supplemental figure. It is also not clear which grounding line was used for this calculation as many are possible according to your Figure 1D.

Line 127: I suggest adding "Kohler Range" to your map so the reader knows exactly where this is relative to the glaciers.

Lines 125-130 and throughout the text: "respectively" should go at the end of the sentence.

Line 128: To focus purely on the glaciers, I suggest removing the clause "to the east of Crosson ice shelf."

Figure 2 caption – see suggestions for figure 1 caption.

Table 1 – I suggest adding which ice shelf the glacier feeds in the "region" column. I suggest bolding the glacier that slowed down (lines 136-137).

Section 3.2: It is not clear why comparing 2005-2022 and 2015-2022? Why is the percentage speed change the best way to report these results? I also suggest adding a sentence about how you calculated the change in speed in the methods or supplemental information. Also, is the rate of speed change from 2005-2022 including the Sentinel 1 velocities or only the measures product?

Line 151: Can you specify what you mean by increased 1.5 times from 2015 to 2022? Also, there is a not a 2005-2015 mean in Table 1.

Line 152: I suggest using a different word than "around" in this context.

Line 155: How are you defining "short term"? Depending on the audience, some might say "from 2005 to 2010 and from 2014 to 2017" are longer term changes.

Figure 3: Overall, this is a nice figure. Can you detail in the methods section how you detrended the velocity? I suggest including in the caption what the yellow shading is. It is difficult to tell which glaciers are which color. Please use different distinct colors for each glacier instead. Previously, you say it is a 17.5 year study period so I am confused here. I also wonder if it I better to use the average of multiple years (say 2005-2008) as a reference year due to the possibility that 2005 could be anomalous in any way.

Line 164: Which panel in Figure 4?

Figure 4: The aesthetic of this figure is very nice! Try to make the other figures more like this one! Please do make the panel letters larger and more in the corner.

Lines 163-165: Is it possible to determine whether the thinning occurred first or that the thinning was a result of the ice flow change? Can you comment on the possibility that the mass input changed or the ice shelf thinning cause a decrease in buttressing stress? It also looks like the thinning is widespread in both the Smith (East and West) and Kohler (East and West) ice streams, according to Figure 1d.

Line 172: There is a new paper by G. Collao-Barrios that discusses a different kind of flow variation that is tidally driven. I suggest including it somewhere in this section. Also, I suggest starting a new paragraph here.

Line 173: Can you elaborate on this sentence and where "elsewhere" is?

Line 174-177: I suggest including this in the discussion and clarifying what you mean. This **has not** occurred on ice caps/glacier scale in a15 year period or it HAS? It is unclear in the current wording.  I also suggest incorporating these references in this portion of the text: Catania et al., 2012 - https://doi.org/10.3189/2012JoG11J219 and Conway et al., 2002 https://doi.org/10.1038/nature01081

Line 178: Please take a look at these references and revise this sentence: Conway et al. 2002; and Price et al, 2001 https://doi.org/10.3189/172756501781832232

Figure 4: This is an interesting plot (4b)! However, the axis label should read 'downstream flux', correct? Smith East seems a bit suspect - is that far upstream increase in flux due the small speed increase? I guess I'm saying, check the numbers for this one specifically, the others look good! Also, are there 'no data' areas in here? What do the pure white patches mean?

Line 190: I'm not sure there is a clear compressive arch for Crosson, or if so, how are you defining it? Do you have strain rate data? I think it's best to just say "inland."

Lines 191-194: This rewording is confusing, consider revising. Additionally, why is the "distance decreasing from the island" of importance?

Line 194: How did you measure the increase in damage?

Figure 5: See general comments.

Line 220: This slow down is difficult to see in Figure 3.

Lines 225-231: This section is very lengthy and needs to be more concise. Can you be more specific what the hydrographic surveys and numerical models show? It is necessary to include the location of PIG on one of your figures so the readers know where the glacier is relative to PSK. PIG also has a very unique geometry compared to PSK (in a relatively narrow embayment compared to Thwaites/DotsonCrosson/etc); is it reasonable to assume the same oceanographic mechanisms are happening there? From 2017 onwards, Kohler East/Smith West Vane/Haynes (Can't tell the color differences) continued to speed up. What would cause this difference between all of these glaciers, is the acceleration different?

Line 238: how does the 82 m/yr compare to your observed data?

Line 240-244: see previous comment

Figure S2: Can you include your method on how you calculated "damage" and how you defined it? In the main text as well as the supplemental information.

Line 246-247: Please include details on how calculated the mass flux? What did you use for bedrock/ice thickness?

Line 245-250: I suggest can marking (maybe on fig. 4) where the ice shelf divide migrated from and by how much it migrated by?

Line 256-258 – The paper you cite suggest that grounding line retreat will resume. "Ice speed did not decrease until 2012 when the cold-water anomaly began. The slowdown was likely also partially a result of the advection of thicker ice onto the bathymetric ridge [Joughin et al., 2016]; the lower ocean heat content likely lowered basal-melt rate, allowing thicker ice to advect farther downstream. As the water in the cavity subsequently warmed, however, speeds increased to their precold anomaly rates, suggesting grounding line retreat will resume." I think the use of "thermocline" is misleading, as it was cold water that entered the cavity below PIG, slowing it less than <4%, then when the warmer water came back the speed went back to accelerating. Though recent modelling studies suggest PIG's GL may be temporarily stable, it's continued thinning and acceleration suggest otherwise. I would not say the Christianson paper agrees with the Hill 2023 paper. They have fundamentally different conclusions. Additionally, to say that none of the glaciers in the PSK region are susceptible to MISI on shorter timescales disregards the recent results by Reed et al., 2024 on the significant GL retreat from 1970-1990 (https://www.nature.com/articles/s41558-023-01887-y).

Line 261: I suggest rephrasing this sentence as the wording is confusing.

Lines 255-268: This part of the discussion needs to be revised to be organized and include clear key takeaways. Please see the suggestions in the general comments.

Morlighem 2017 is not in the reference list.

Equation S1 – please format the units correctly, like you did for the others:

A = rate factor for ice (9.3e-25; s^-1 Pa^-3)

---

## Author Comment (AC1)

**Response to Reviewer Comments - Speed-up, slowdown, and redirection of ice flow on neighbouring ice streams in the Pope, Smith and Kohler region of West Antarctica**

We thank the reviewers for their time and effort in reviewing our paper, "Speed-up, slowdown, and redirection of ice flow on neighbouring ice streams in the Pope, Smith and Kohler region of West Antarctica", submitted for publication in the cryosphere. We welcome the positive feedback and insightful comments which we have endeavoured to fully address in this resubmitted revision, and we hope you agree this improves the manuscript. We have incorporated the majority of the suggestions made, and in the limited cases where we have not, we have provided a detailed description of the justification foreach decision. The changes are highlighted in the manuscript. Please see below a point-by-point response to the reviewers' comments, where all line numbers refer to the revised manuscript file.

**Reviewer 1: Felicity McCormack**

This study characterises changes in ice flow in the Pope, Smith, and Kohler region of West Antarctica from 2005-2022 using satellite observations from Sentinel-a and 1b combined with MEaSUREs estimates. The manuscript is well written, aims clear, and the analysis well executed. I don't have any major issues, and I suspect the points that I've highlighted below should be relatively straight forward to address.

**Reviewer 2:**

Selley et al. document a velocity change in 8 glaciers that feed the Dotson and Crosson ice shelves over a recent 17.5-year period. They found most of the glaciers accelerated, with the exception of one tributary, Kohler West, which slowed down. This resulted in a change in the location of the ice divide separating the two ice shelves. The authors infer one step further and suggest the change in the mass flux into the two ice shelves contributed to the disintegration of part of the Crosson ice shelf.

The Amundsen Sea Embayment is a "hot topic" for Cryospheric studies due to the warming of the region and the sea level rise potential of the outlet glaciers. This study contributes unique findings on the less studied glaciers of the region (relative to Pine Island Glacier and Thwaites). However, some issues need to be addressed prior to publication. Please see my comments below.

| Line | Comment | Response |
|---|---|---|
| **Reviewer #1** | | |
| 1 | My main issue is that I came away a bit murky about what's driving what changes in this region and how that relates to instability (this could be partly because I'm less familiar with the history of this region!). For example, the Crosson Ice Shelf has had some very large ice front retreat | Comment. We thank the reviewer for their feedback and agree it is difficult to distinguish the sequence of changes and causation. It was out of the scope of this study to do an expansive analysis of all drivers, but we hope that |

| | | |
|---|---|---|
| | associated with large calving events, the margin between Dotson and Crosson has migrated eastwards, and there has been an increase of flux into Crosson. But I found the discussions around the drivers of these changes (i.e. the sequence of changes and causation) and how they relate to the overall stability of the system a bit unclear. Stepping through these aspects in the discussion in a bit more detail for both Crosson and Dotson (i.e. in paragraphs starting on L233, L245 and L255) would be helpful. | the detailed edits below, in particular those relating to the discussion sections of the paper have addressed the reviewers' concerns. We agree with the reviewer that identifying the chronology of changes would be of great benefit and hopefully this study reports the observed changes and a basis for further detailed investigation of individual drivers. |
| 2 | L30 "While Ice loss"→ "While ice loss" | Done.

**Edit L33:** "While ice loss" |
| 3 | Section 2.1. Could you add a couple of sentences about the Sentinel data, including the overall timeframe of the data, resolution, and image pairs considered. | Comment. We thank the reviewer for their comment about the level of detail in our methods. It's always a tricky balance to provide an appropriate level of detail. While reviewer two has requested slightly less detail, or that the information be relocated into the supplementary material, reviewer 1 requested slightly more detail. Given these opposite requests we have elected to keep the level of detail as it is because this is our preference, but it also seems to be the middle ground between both reviewers. |
| 4 | L78 Figure S3. It'd be great to get subtitles on each figure panel and to include a 3rd column that shows the % differences in flow speed. It's otherwise hard to eyeball a 30% error in the shear margins. | Done. We have added percentage error panels to S3 and updated the figure to match Reviewer 2's suggestions for improved formatting.

**Edit Figure S3:** |

[Figure]

[Figure]

| 5 | L81-100 **Removal of data points.** It would be helpful for a non-expert for a few extra details here:
• What % of the data were removed (including outliers etc)?
• Out of interest, are these points somewhat randomly distributed over the whole study region or do they concentrate?
• L85: Some words about why the values of 5.8 SNR and 45 degrees were chosen would be helpful.
• Is the approach that you've taken standard (asking as a non-expert)?
• It's not essential, but I wonder if a schematic of the workflow could be | Done.

Most satellite datasets require some pre- and post-processing filtering to ensure erroneous values aren't used. This is not just specific to satellite datasets, many in-situ or airborne datasets will also be filtered at some stage. The approach we use directly builds on the method used in a number of previous publications (Lemos et al., 2018b, a; Selley et al., 2021; Surawy-Stepney et al., 2023; Wallis et al., 2023), and to ensure full transparency we have set out the steps taken in this paper also so that the same approach |

| | | helpful to readers who aren't very familiar with the procedures | could be easily used by other researchers.

With regards to the percentage of data removed, this number will vary from image to image depending on how good quality the velocity tracking output is for that pair. (Note, in our error section we explain what types of condition cause errors to occur, e.g. snow melt or snow fall which can affect the amplitude of the SAR image, and therefore makes recognition of the same features difficult, as is required for IV tracking method.) Some data is also removed/filtered at the ice velocity tracking step which is why you have spatially variable data gaps visible in the speed maps. The key point is that the thresholds used for removing any outliers remain constant at all stages of the processing chain, so the method used for this is consistent and repeatable.

Regarding the locations of the erroneous points, it would be impossible to predict where they will occur, but they aren't randomly distributed either. We tend to find erroneous points clustered around the highly crevassed areas such as the shear margins (these regions deform as well as displace so it is hard to track the same features between image pairs, even though it is just a 6 or 12-day repeat window), but also in the slower moving interior where there is a paucity of visible features to track, such as crevasses.

We have included new % error maps, see Supplementary figure 3.

We haven't provided a workflow of the ice velocity processing chain as it is a mature method, and as this paper isn't significantly altering that method, we don't think it would add value. We would be very happy to provide a workflow to anyone that requests it if |
|---|---|---|---|

| | | they get in touch with the corresponding author. We have a slide on this that we use when giving conference presentations, and it was also a figure in my thesis which is publicly available. |
|---|---|---|
| 6 | Can you make the supplementary figure references chronologically ordered in the manuscript? Currently fig. S3 comes before S1 and S2 and table S2 before S1 | Done.

**Edit Figure S2:** Reordered supplementary figures.

**Edit L78:** Fig. S2
**Edit L105:** Fig. S2
**Edit L192:** Fig. S3 |
| 7 | 101-105 Is there good agreement between MEaSUREs velocities and those estimated here using Sentinel for the overlap time period (2015-2017)? A comment on this, or perhaps a few sentences in the supplementary information, would be helpful. | Done.

The Sentinel-1 and MeASUREs data overlap between 2016 and 2017. We compared the two datasets in this time period in the 2.5 km diameter regions at the grounding line of all 8 glaciers, see Table below. Overall, the Sentinel-1 velocity measurements are slightly faster than MeASUREs result on all glaciers, with an average speed difference of 21 m/yr (5%) in 2016 and 17 m/yr (3 %) in 2017. If we remove the slower flowing Horrall and Vane glaciers which flow at 401 and 203 m/yr respectively, the absolute difference reduces slightly (19 and 13 m/yr respectively), but the percentage difference reduces substantially to 2 and 1 % for 2016 and 2017 respectively. This is well within the error on our speed measurements.

The majority of this speed difference is likely caused by differences in the underlying spatial resolution of the satellite data and the step and window size used for the feature tracking. It is well known (Lemos et al., 2018a) that finer spatial resolution satellite datasets allow you to track ice speeds at high spatial resolution which then detect small regions of fast flow. Equally using a larger window and step size in the feature tracking step will tend to effectively smooth the output |

ice speed result which subtly reduces the average mean speed, and it also tends to increase spatial coverage slightly. As we move into an era with more SAR satellites that enable us to track ice speed at different resolution from different sensors for any one location, it will be increasingly important to characterise these differences, in the same way the satellite altimetry community is already doing for laser and multi-frequency radar altimetry products.

There is also a slight difference in the time periods covered by the two products: Sentinel runs Jan-Dec whereas MeASUREs runs from July to June. We know from IMBIE studies that any difference in the spatial and temporal domain of different datasets can cause differences between them. These differences may be due to error int he products, however they may also be due to real geophysical change that occurs between the time periods. We used a linear fit in all ice speed trend analyses to minimise the impact of any offset between the two speed products.

| Time Period | Horrall | Kohler West | Kohler East | Smith West | Smith East | Pope | Vane | Haynes | Average (all) | Average (fast flowing) |
|---|---|---|---|---|---|---|---|---|---|---|
| | | | | MeASUREs minus S1 (m/yr) | | | | | | |
| 2016 | -17 | -18 | -28 | -8 | -28 | -14 | -42 | -15 | -21 | -19 |
| 2017 | -33 | -3 | -28 | -12 | -6 | -2 | -21 | -29 | -17 | -13 |
| | | | | Average S1 Speed (m/yr) | | | | | | |
| 2022 | 401 | 715 | 1215 | 1188 | 1093 | 772 | 203 | 810 | 800 | 966 |
| | | | | Difference between Measures and S1 speeds as % of 2022 speed | | | | | | |
| 2016 | -4 | -3 | -2 | -1 | -3 | -2 | -20 | -2 | -5 | -2 |
| 2017 | -8 | 0 | -2 | -1 | -1 | 0 | -10 | -4 | -3 | -1 |

| 8 | L117: Is Fig. 1e missing or is this meant to reference a different figure? | Done.

**Edit:** Removed figure reference. |
|---|---|---|
| 9 | Figure 2, panels c-j: It's difficult to orient these lines and to know here they start and where they finish. What does 0 km (x-axis) represent? It would be helpful to be able to compare with figure 3 from Milillo et al. (2022) re the discussion on the importance of the prograde slope of Kohler for the stability of this ice stream (ref L234-236). Also, are the ice speed tick labels on the left y-axes of panels c-j correct? The speeds are low. | Done.

We edited the figure caption, corrected the units and re formatted as per reviewer 2's suggestion. Please see below for revised figure. |

**Edit Figure 2:**

**Edit Figure 2 Caption**: "The x-axis is shown as distance from the grounding line, with positive values indicating the inland section of the profile on the ice sheet, and negative values indicating seaward locations."

| 10 | Figure 3: what are the shaded regions in figure 3a? | Done.

**Edit Figure 3 Caption:** "The shaded regions represent the rapid periods of speed up demonstrating the periodicity around the 17.5-year trend." |
|----|-----------------------------------------------------|------------------------------------------------------------------------------------------------------------------------------------------------------------------------------------------------------------------------------------------|
| 11 | L163 By what degree do the ice flow vectors rotate in the piracy from KW to KE? From figure 4 it looks like it's mainly restricted to <\|30 degrees\|, but there are some regions where the colour bar saturates in figure 4d. I'm thinking back to the removal of ice velocities that have > 45-degree rotation and wondering if data from some pixels could be removed inadvertently? | Done.

We thank the reviewer for highlighting this could be more clearly shown. The 45-degree rotation filtering was applied at image pair level not on the annual mosaics, i.e. 6–12-day interval pairs where we are confident that level of rotation is an error. We investigated the areas of highest directional change the Table S1, where the maximum rotation for any year was 39 degrees. We have added the rotation values to the text and more clearly stated the ROI in S1 relate directly to Figure 4c and d. |

| Time | ROI 1 | | ROI 2 | |
|---|---|---|---|---|
| | Flow Direction (degrees east of north) | Error (degrees) | Flow Direction (degrees east of north) | Error (degrees) |
| '30-Dec-1996' | 25.06 | 0.09 | 30.19 | 0.18 |
| '30-Dec-2000' | 26.45 | 0.26 | 23.90 | 0.69 |
| '30-Dec-2005' | 26.33 | 0.11 | 25.94 | 0.21 |
| '30-Dec-2006' | 26.42 | 0.21 | 25.54 | 0.45 |
| '30-Dec-2007' | 25.30 | 0.12 | 26.74 | 0.21 |
| '30-Dec-2008' | 25.39 | 0.16 | 25.51 | 0.32 |
| '30-Dec-2009' | 25.89 | 0.13 | 28.31 | 0.20 |
| '30-Dec-2010' | 25.70 | 0.11 | 30.73 | 0.19 |
| '30-Dec-2011' | 25.72 | 0.61 | 31.85 | 1.35 |
| '30-Dec-2012' | 23.39 | 0.83 | 32.47 | 1.78 |
| '30-Dec-2013' | 25.33 | 0.13 | 34.64 | 0.62 |
| '30-Dec-2014' | 24.80 | 0.23 | 34.60 | 0.67 |
| '30-Dec-2015' | 24.54 | 0.44 | 34.50 | 0.86 |
| '30-Dec-2016' | 24.04 | 0.16 | 35.01 | 0.36 |
| '30-Dec-2017' | 23.34 | 0.07 | 37.00 | 0.13 |
| '30-Dec-2018' | 23.10 | 0.07 | 37.82 | 0.13 |
| '30-Dec-2019' | 22.79 | 0.06 | 38.41 | 0.12 |

**Edit Line 171:** "eastward rotation (~8⁰) of the ice flow vectors by 2019"

**Edit Line 175:** "westward rotation of the flow vectors (~2⁰)"

**Edit Table S1 Caption:** "Summary of flow direction changes in the Regions of Interest (ROI) identified in Figure 4 based on ice speed and flow direction data (ROI1 referring to the region shown in Figure 4c and RO2 referring to the Figure 4d region. This clearly has been a progressive change in flow direction over time, particularly since 2009 at ROI2."

| 12 | Figure 4b: It would be helpful to see the change in thickness over the same period so that it's clear where flux changes are due to flow piracy vs dynamic thinning. There are conflicting labels: "downstream flux" in the caption, but "upstream flux" in the colour axis label. | Done. We have provided further detail on the flux calculation. The upstream flux was calculated by seeding flowlines in 2005 and 2019, then at every 100 m along all flow lines the discharge calculated through that point. It was then integrated 'up' along each flowline and gridded for plotting and differencing.

This metric is intended to demonstrate how a change in speed or flow |

| | | direction at the division of Kohler East and Kohler West impacts the mass flux into the downstream ice shelves. They show that less ice going from Kohler West into Dotson, but more going from Kohler East into Dotson, at the expense of Crosson. |
| --- | --- | --- |
| | | A static thickness for both timestamps was used as the plot was purely exploring the impact of speed and flow direction change. We have amended the text to more clearly reflect this. |
| | | **Edit figure 4 caption:** "upstream flux in 2019". |
| | | **Edit L105**: To establish how the change in flow direction impacts the ice flux into the ice shelves the upstream flux was calculated by seeding flowlines from the calving front in 2005 and 2019, then the flux calculated at every 100 m along each flowline. It was then integrated along each flowline and gridded for plotting and differencing. Velocity vectors were rotated parallel to the flowline to capture the mass flux every 100 m. A static thickness (Fretwell et al., 2013) was used to explore purely the impact of ice speed and flow direction changes. |
| 13 | Figure 5a: re the colours of the calving front contours in the Crosson Ice Shelf – was there retreat and then readvance to near the original calving front position in the latter part of the time period? It's a little bit difficult to date the different positions with the colour map used, and I'm uncertain whether the calving front has continued to retreat over this period. This would be helpful to know and to be able to visualise as it could help with the interpretation of the post-2014 behaviour in figure 3. Perhaps a zoom in of the calving front position here could be helpful, and some indication of the timing of retreat/readvance with labels. Also, are the figures in vector format? I couldn't zoom in to see the details. | Done. We have used a different colour map however the eastern side of Crosson has a very complex calving front with rapid changes making it extremely difficult to show the full detail of the changes. We have added a zoom panel of Crosson and a short description of the post 2014 behaviour. **Edit L191:** After the 2014 calving event, the shape of the shelf remains convex and in a fairly consistent configuration, with small further inland penetration, for example in 2016 (~10km further at the maximum extent). This positioning |

| | | is maintained with additional fracturing and rifting gradually penetrating further across to the western side of the shelf. Notably, in 2021 there is an advance in the floating ice in front of Haynes Glacier which partially recloses the open pathway of water accessing the eastern side of Crosson. In 2022, this configuration persists however the shear margin in front of Mount Murphy fractures and partially detaches.

**Edit Figure 5:**

**Edit:** All spatial figures have been exported at a high DPI. |
|---|---|---|
| 14 | L217-219 I'm not sure I understand the sentence, particularly the references to "interrupted" on L219. | Done. We have amended the sentence to improve clarity.

**Edit L220:** "We note periods of rapid speed-up 2005 to 2011 with an average speed difference across all ice streams of 14 % and during 2014-2017 (12 %) were interrupted by periods of slow down between 2011-2013 (4 %) and 2017-2020 (2 %) (Fig. 3a); the timing and relative magnitude of these speed fluctuations vary between ice streams, but the absolute magnitude of the speed change is similar across the region (Fig. 1)." |
| 15 | L216-231 It would be good to link the variability post-2014 of the ice streams that feed Dotson to the calving front position in this discussion, particularly if it's possible to attribute (at least qualitatively!) changes in the velocities to calving and ocean forcing separately. I would have imagined that the retreat of the calving front into the compressive arch region could have caused more marked speedup (I'm assuming that there has been some more ice front retreat after 2014), but that is | Done. We have added a description of post 2014 Crosson calving behaviour and zoom panel to Figure 5. We have also added text exploring the impact of ice advancing in front of Haynes potentially limiting the access of warm water and the potential re-buttressing of Crosson. We agree with the reviewer that this is an interesting discussion point. We didn't include it in the earlier version of the paper as we wanted to |

| | | |
|---|---|---|
| | Not reflected in the trends (figure 3) where the PSK rates of speedup have decreased (generally, although that wasn't the case for Horrall). Why do you think this is the case? Or do the speed changes post-2014 link directly to the thermocline variability (here, a decrease in melt rates)? | avoid speculating too much but we welcome the opportunity to elaborate in more detail on these points.

**Edit L234:** "Particularly, the rate of speedup decreasing after 2020 may also be linked to the advance in the floating ice in front of Haynes Glacier which partially recloses the pathway of open water accessing the eastern side of the Crosson Ice Shelf perhaps limiting the penetration of the warmer water into the cavity. A re-buttressing of the Crosson Ice Shelf may have also occurred, albeit in a new configuration, with the calving front connecting to Mount Murphy as a pinning point and lowering the rate of observed speed up."

**Edit L193:** After the 2014 calving event, the shape of the shelf remains convex and in a fairly consistent configuration, with small further inland penetration, for example in 2016 (~10km further). This positioning is maintained with additional fracturing and rifting gradually penetrating further across to the western side of the shelf. Notably, in 2021 there is an advance in the floating ice in front of Haynes Glacier which partially recloses the open pathway of water accessing the eastern side of the Crosson Ice Shelf. In 2022, this configuration persists however the shear margin in front of Mount Murphy fractures and partially detaches. |
| 16 | L249-254 What do you mean by the divide migration between Dotson and Crosson having maintained the stability of the Dotson Ice Shelf? Also, it'd be good to elaborate on how the redirection of ice from Kohler East has exacerbated Crosson deterioration. | We have aimed to improve our description here. The rotation of ice flow and flux ROI 2 (Figure 4d) has offset the rotation of flow in ROI1 (Figure 4c) as such the sustained flux into Dotson have increased and maintained the ice feeding Crosson therefore exacerbating the deterioration as the reviewer suggests.

**Edit L264:** "However, the discharge from Kohler East Glacier into the eastern part of Dotson Ice Shelf |

| | | increased by 5.6 Gt (26 %) over the same time period because of the acceleration of Kohler East Glacier and redirection of ice flow resulting in an eastward migration of the ice divide between Dotson and Crosson Ice Shelves." |
|---|---|---|
| **Reviewer #2** | | |
| 17 | Section 2.1 is highly detailed. It is refreshing to see the intricacies of the analysis; however, it is not necessary to include everything described in the main text. I suggest including the specifics in the supplemental information and keeping the main text clear and concise. | Comment.

We thank the reviewer for their positive comment about the level of detail in our methods. It's always a tricky balance to provide an appropriate level of detail. While reviewer two has requested slightly less detail, or that the information be relocated into the supplementary material, reviewer 1 requested slightly more detail. Given these opposite requests we have elected to keep the level of detail as it is because this is our preference, but it also seems to be the middle ground between both reviewers. |
| 18 | Section 3.3 the possibility of "ice piracy" and its relationship to other "ice piracy" events. This is a fascinating topic and I believe this would fit much better in the discussion. Most past papers on ice piracy have invoked changing subglacial hydrology, but I think using the term here is fine. Notes posted in the marked-up text suggest a few additional references. | Done. We have moved the information on ice piracy to the discussion section of the paper.

**Edit L248:** These changes in flow direction are consistent with the observed changes in surface elevation and ice thickness in this region and would be expected to occur elsewhere in places where the geometry permits. Analogous historical changes in flow direction and ice divide migration have been inferred from chemical tracers (Iizuka et al., 2010) and are widely documented in paleo ice sheet reconstructions (Conway and Rasmussen, 2009; Iizuka et al., 2010), whilst ongoing ice divide migration has been inferred from isochrones (Nereson et al., 1998) and thickness change measurements (Conway and Rasmussen, 2009). To our knowledge, however, redirection of ice flow from one ice stream to another has not been observed directly on ~15-year timescales, it may have occurred at |

| | | glacier and ice cap scale. Differential thinning near ice stream boundaries is not isolated to this study region; therefore, we expect that ice divide migration and the associated changes in flow direction have and will occur elsewhere in Antarctica, and this should be considered when interpreting observed changes in ice speed and ice thickness in tightly connected regions. Modelling work indicates that minor changes in ice sheet geometry can result in 'ice piracy' and highlights the importance of quantifying the rates of change and subsequent impact on ice mass loss (McCormack et al., 2023). |
|---|---|---|
| 19 | The discussion requires more coherent organization and clearer key take-aways. There were many aspects of the results that felt thrown into the discussion (e.g., Lines 218-220 – this observation should be mentioned in Results first). I suggest reorganizing the paragraphs into three main sections: Long-term trends – where you compare to PIG and MISI; Short-term periodicity – where you discuss the variability in the velocity on the annual scale and potential causes; and the connection to Dotson/Crosson – the "ice piracy" event, movement of the ice divide, and the subsequent deterioration of Crosson. It would also be interesting to see more information in the discussion about the ice shelf ice front and the rift propagation, and any connection between those results and the velocity results. | Comment. We apologise if the reviewer felt the discussion wasn't clear. We spent quite some time writing what is quite a complex narrative, and we hope this comes across in the overall quality of our writing. We have formatted the discussion to discuss the following topics in order:

 • Trend agreement with previous work
 • Short term periodicity
 • Kohler West slow down.
 • Dotson & Crosson ice flux & stability
 • MISI

 We hope this clarifies the structure to the reader, and we do feel that it is the most sensible way to order this information which is why we have not chosen to revise the order. |
| 20 | A quick note on the short-term periodicity – could it be driven by ungrounding of the numerous ice rises in the Crosson-Dotson (especially the latter) system? It is important to entertain the other possibilities that it is not related directly to the ocean dynamics – without more oceanographic observational data in this local region and time period. It seems far more likely to me that the loss of the pinning points, indirectly related to warm ocean conditions at depth (and not necessary linked to | Done. We agree that the loss of pinning points would also impact ice speed. We observe the disintegration of the eastern side of the Crosson ice shelf in 2014 and concave calving front configuration persists through to 2022. There is potential that Mount Murphy on the eastern side of Crosson has become a stabilising pinning point after the initial speed up observed and contributed to reduced rate, we see post ~2019. Although, the new |

| | | |
|---|---|---|
| | ocean circulation events at PIG) are the cause of the rapid speed-up periods. | configuration of Crosson is still conducive to easier penetration of the warm water in the cavity. There are two ice rises identified in BedMachine in front of Kohler East, whether theses unpin is underknown, additionally whether the cavities under Crosson and Dotson connect is also unknown. We have added the following text to include this hypothesis in the discussion.

**Edit L238:** Another contributing factor could be the ungrounding and re-pining of ice rises beneath the ice shelves. Notably, there are two rises directly downstream from Kohler East and Smith West, and should their pinning strength change, this would result in a significant speed change. It also remains unclear whether there is a pathway for the ocean to flow between the Crosson and Dotson Ice Shelves, and how this pathway has evolved historically. |
| 21 | Regarding the very interesting "ice piracy" event - it would be beneficial to include more information about the thinning that occurred specifically during the period of the ice flow change. Additionally, it would be advisable to discuss the widespread thinning throughout the region and why the diversion of flow did not happen there. | Done. We agree with the reviewer that this is an interesting point. We propose it's the juxtaposition of the slower thinning Kohler West (-1.1 m/yr) and faster Kohler East (-2.5 m/yr) which has driven the ice piracy not just the thinning. Whilst Smith West and Smith East thin at similar rates to Kohler East (~2.5 m/yr) their geometry does not result in the spatial gradient of thinning we see at Kohler West. As the focus of our study was to investigate and measure the changes in speed and its direction, we think that it would be reaelly interesting for a future study to focus on the detailed ice thickness changes to understand their size in more detail, but also the timing of any change in the rate of ice thickness change.

**Edit L273**: "We do not observe ice piracy occurring in any of the other ice streams in the region (Figure 4). This is likely due to their geometry which does |

| | | not result in the spatial gradient of thinning we observe at Kohler West/East junction." |
|---|---|---|
| 22 | I also found the font to be blurry on most of the figures, I suggest making sure the plots are of a higher DPI (300 minimum), especially in the supplemental figures. | Done. **Edit:** All figures have been exported at a DPI higher than 300. |
| 23 | The figures in the text show a lot of important information but are a bit hard to follow in some cases. For all of the figures that show spatial data it would be better to move the legend scalebar into each respective panel -- following which scale bar belongs to which panel takes significant effort. | Done. We have endeavoured to implement most of their suggestions where possible. **Edit Figure 1, 2 & 5.** |
| 24 | Figure 1: I suggest making each panel larger to fit more information on the panel itself. For example, each panel needs a label and legend. I suggest moving the profile line labels/glacier names so that they are more clearly associated with each profile line. I'm not sure the 'bed topography data is very useful – it does not show up under any ice-covered area (where it would be more interesting – either it should have its own panel or perhaps just remove it. If you keep it, re-consider the scale – perhaps just -3000 to 0 meters is adequate? The 0-25km scale should be located on one of the panels. I suggest including an Antarctica overview map in one of the inset panels (preferably with ASE and PIG labelled since you discuss these in detail). The period of the "rate of speed change" should be labelled on the panels | Done. We thank the reviewer for their advice on improving the clarity of the figure, we have endeavoured to implement most of their suggestions. We have enlarged the panels, repositioned the glacier names, removed bed topography from the three-ice speed related panels and adjusted the colour scale for the bed topography in panel d. **Edit Figure 1:**  |
| 25 | Figure 2: I suggest labeling the profile in 2b with 'Grounding Line profile', and also increase the size of the font of the glacier names and center them on the panels. You can gain space for your graphic panels by putting the labelling letter inside the panel (e.g. bottom left corner). The ice speed legend could be inside panel A. Panels C-J could be expanded to be the same size as the upper two panels (see below – red arrow) and tighten the spacing between panels C-J. Make the "distance along gate" legend larger | Done. |

| | | |
|---|---|---|
| | (now that the scale bars are moved), and change "gate" to profile. In legend, "surface" and "bed" could be on the panels themselves to make the figure space more efficiently used | **Edit Figure 2:**
 |
| 26 | Figure 5 – The calving front is difficult to distinguish for Crosson, especially in the north west corner. I suggest a zoomed in inset. The scale bar legends should be placed into the panels and Panel B made larger. The rift time series could be labeled with the year instead of the color bar (It would be clearer this way and since there are only 7 years and it won't take up too much space – especially when you make the panel larger). Scale bar is too small – make a bit longer and less bold (0-50 km? with ticks instead of a solid black square). I suggest labelling Bear Island on both panels. Place panel label "B" in top right corner to match panel A. Specify panel B is the orange rectangle in panel A. | Done.

**Edit Figure 5:**
 |
| 27 | Figure S1 – Please see the comment about the DPI quality of the figure. I also suggesting aligning the legend scales with the respective panels. I am wondering how panels H-J are different than Figure 4? | Done. We have removed the duplicated panels and exported at a higher API.

**Edit Figure S1:**
 |
| 28 | There are more significant figures than can be justified - rounding to the nearest 10th for the error and velocity measurement would be better. | Done.

**Edit L144:** ice speeds increased by 32 m/yr$^2$ from 2005 to 2022 |

| | | |
|---|---|---|
| | | **Edit L145:** Haynes (17 m/yr$^2$) and Smith East (14 m/yr$^2$)

**Edit L154:** decelerated by -5 m/yr² since 2005 |
| 29 | Line 26: I suggest including a definition of MISI. | Done.

**Edit L26:** "MISI is theorised to occur on marine ice sheets and is an instability where the ice is grounded below sea level on bed rock that slopes downwards into the interior of the ice sheet. This configuration has the potential to cause rapid retreat of the grounding line and increase ice flow into the ocean (Schoof, 2007)(." |
| 30 | Lines 30-36: I think it is better to introduce dynamic details after the introduction of geography of the area. See comment in general comments. Also, A newer paper by Clark et al., 2024 provides further support for the 1940s kick-off for ice loss https://doi.org/10.1073/pnas.2211711120 | We thank the reviewer for their suggestion however we feel the context of the Amundsen Sea Embayment and the driving factors of the change is important to set the background before focusing on the PSK region. We have included the recent paper reference. Thank you for this suggestion.

**Edit L32:** "Clark et al., 2024" |
| 31 | Lines 41-42: It would be better to include this before line 36 before talking about Kohler dynamics. Throughout this paragraph I suggest to only use East and West Glacier Name instead of alternating between East/West/Singular Glacier Name.

It is unclear which tributary of the glacier stream you are referring to when you only use the singular name. | Done. See previous comment response (30). We have amended the glacier names as requested.

**Edit L34:** "(PSK) Ice Streams" |
| 32 | Lines 42-43: I suggest adding rifting detail "ice shelf has extensive rifting, particularly on its eastern side" after line 43-45. | Comment. Apologies, we don't fully understand the comment so haven't been able to address it. We will be happy to address it if a clarification could be provided? |
| 33 | Line 45: I suggest revising this sentence "Kohler West Glacier and the Dotson Ice Shelf into which it flows have changed less over the last 30 years" to – "Kohler West Glacier, which flows into the Dotson ice Shelf, have changed less | Done.

**Edit L48:** Kohler West Glacier, which flows into the Dotson ice Shelf, has changed less over the last… attributed to the prograde slope." |

| | | |
|---|---|---|
| | over the last… attributed to the prograde slope." | |
| 34 | Lines 45-50: I suggest revising these sentences as the structure is a bit confusing. | Done.

**Edit L48:** Kohler West Glacier, which flows into the Dotson Ice Shelf, has changed less over the last 30 years, attributed to its prograde bed slope (Milillo et al., 2022; Scheuchl et al., 2016). Satellite observations show that the grounding line of Kohler West Glacier has retreated 200 m/yr between 1992 to 2011 (Milillo et al., 2022), with no change in ice speed observed up to 2015 (Scheuchl et al., 2016) despite the ice shelf thinning near the grounding line (Gourmelen et al., 2017; Zinck et al., 2023). |
| 35 | Line 51: I think you mean this '...speed change since 2015 is not well characterized, and....? | Done.

**Edit Line 53:** "the spatial pattern of its speed change since 2015 is not well characterised, and…" |
| 36 | Line 55: for 17.5 years from 2005-2022. | Done.

**Edit Line 56:** "for 17.5 years from 2005-2022" |
| 37 | Figure 1 caption – I suggest shortening the figure caption. For example, "Average 2015 to 2022 ice speed over the PSK region, measured using interferometric Wide (IW) mode synthetic aperture radar (SAR) data acquired by the Sentinel-1a/b satellites." It would be sufficient to say "Sentinel 1 SAR derived averaged ice speeds over PSK region from 2015-2022."

It is also not mentioned where the velocities from 2005-2015 come from – I thought they were from 2005-2022? It alternates throughout the text, please be consistent.

Additionally, please write the dates with backslashes or dashes instead of periods. The periods make it look like a number and not a date. Please fix this throughout the text. As a quick note, if you include the method of measurement for one panel in the caption please then do so for the others – however, I suggest not including it in the caption and reserve the details for the manuscript text. | Done. We have amended the caption, date format and the time periods throughout.

**Edit Figure 1 Caption:** Ice speed and rate of speed change at Haynes, Vane, Pope, Smith East, Smith West, Kohler East, Kohler West and Horrall Glaciers, which feed the Crosson and Dotson Ice Shelves in the Amundsen Sea Sector of West Antarctica. (A) Sentinel-1 SAR derived average speed 2015 -2022. The 2011 grounding line location (solid black line) (Rignot et al., 2016) and the location of the 8 flow line profiles (dashed black lines) are also shown. **(B)** Observed rate of speed change over the full 15-year study period, from 2005/05/01 to 2022/05/30. **(C)** Observed rate of speed change during the Sentinel-1 period from 2015/06/01 to 2020/05/31. Measurements are |

| | | superimposed on BedMachine bedrock topography (Morlighem et al., 2017). **(D)** Rate of elevation change for 1992 to 2023 (red) (Shepherd et al., 2019) and grounding line locations from 1992 to 2020 (blue) (Milillo et al., 2022; Rignot et al., 2016).

**Edit on Dates:**
• L103: 01/06/2005 to 31/05/2017 |
|---|---|---|
| 38 | Line 69: "which" should have a comma beforehand | Done.

**Edit Line 68:** "technique, which" |
| 39 | Line 70: Here you list features and then say "stable amplitude variations", which is not a feature. | Done.

**Edit Line 69:** "visible features at or near to the ice surface such as crevasses or rifts and stable amplitude variations" |
| 40 | Lines 72-74: I suggest including this in the supplemental information. | Comment. As mentioned in previous responses, the requests from both reviewers 1 and 2 are different here, so we have opted to keep the methods section as it is as this is the middle ground. We have kept the level of detail specified in line with previous publications (Lemos et al., 2018b, a; Selley et al., 2021; Surawy-Stepney et al., 2023; Wallis et al., 2023). |
| 41 | Line 76: I suggest changing to "...and signal propagation speeds in the ionosphere… etc" | Done.

**Edit Line 75:** "and signal propagation speeds in the ionosphere" |
| 42 | Lines 78-79: please report the lowest error and average error.

Additionally, Figure S2 was not referenced before, so I suggest changing S3 to S2 and correcting the text and supplemental information to have the correct figure numbers. | Done. We have reported the maximum error in the main text in order to provide the most conservative estimate on the quality of our data. We have also ow included the error maps which vary in space and time which provide a detailed overview of the speed error across the region.

**Edit Figure S2:** Reordered supplementary figures.

**Edit L77:** Fig. S2
**Edit L105:** Fig. S2
**Edit L190:** Fig. S3 |

| | | |
|---|---|---|
| 43 | Figure S3: Please clean up the figure and follow suggestions made in Figure 1 and general comments (e.g. legends, scale, etc. also, the lat/lon does not need to have the minutes or seconds included as they are all 0. Keep it simple and clear). | We thank the reviewer for their suggestions to improve our figures, we have implemented their suggestions where possible.

**Edit Figure S3**: Please see response to comment 4. |
| 44 | Line 83: I am wondering which reference speed map is this? How is it different than the MEASURES data also used? I suggest restructuring the sentence to include "First" so it matches the rest of the paragraph. | Done.

**Edit L82:** "first we compare each speed field to the nearest MeASUREs reference speed map" |
| 45 | Line 84: I suggest removing passive voice (ie. rewrite as: "we remove..etc") | Done.

**Edit Line 83:** "Secondly, we removed velocity" |
| 46 | Line 104: See comment about dates | Done.

**Edit Line L103**: "01/06/2005 to 31/05/2017" |
| 47 | Lines 110-112: Can you comment on the limitations of only using imagery from 1-2 months of the summer season? | Done.

**Edit L113:** Images were acquired from mid-January to the end of February to ensure consistent temporal sampling, to use relatively cloud free data, and to avoid aliasing seasonal variation. The calving fronts were manually delineated (Cook et al., 2005; Cook and Vaughan, 2010), and the annual resolution allows an overview of the calving front changes on the 17.5-year timescale. |
| 48 | Line 117: There is not a Figure 1E. | Done.

**Edit:** Figure reference removed**.** |
| 49 | Line 121: You use both flow units and ice streams here. What delineates a "flow unit" if not an ice stream? Please use consistent terminology.

Additionally, what is meant by ice streams penetrating 75 km inland of grounding line? What happens after that? | Done.

**Edit L78:** "major ice streams"
**Edit L126:** "8 major ice streams"

**Edit L127:** ", with the fast flowing regions of the ice streams extending up to 75km inland of the grounding line." |
| 50 | Line 125: Which part of the time series was used for the averaging? | Done. We have amended the text and figure in line with this comment. |

| | | |
|---|---|---|
| | I also suggest putting these circles on Figure 1 or in a supplemental figure.

It is also not clear which grounding line was used for this calculation as many are possible according to your Figure 1D. | **Edit L131:** "speeds up to 1,215 ± 275 m/yr in 2022 "
**Edit L135:** "810 ± 169 m/yr in 2022"

**Edit Figure 2:**

Edit Figure Caption 2: "The 2011 grounding line location (solid black line) (Rignot et al., 2016), the location of the 8 flow line profiles (dashed black lines) and the intersection 2.5 km buffer (purple circles) are also shown."

**Edit L130:** "flow lines intersect with the grounding line (Rignot et al., 2016)." |
| 51 | Line 127: I suggest adding "Kohler Range" to your map so the reader knows exactly where this is relative to the glaciers. | Done.

**Edit Figure 1:**
 |
| 52 | Lines 125-130 and throughout the text: "respectively" should go at the end of the sentence. | Done throughout text.

**Edit Line 48:** "Both Dotson and Crosson Ice Shelves have thinned (Pritchard et al., 2012) by 10 % and 18 % (Paolo et al., 2015) between 1994 and 2012, with |

average basal melt rates of 5.4 ± 1.6 and 7.8 ± 1.8 m/yr (Adusumilli et al., 2020), respectively."

**Edit Line 127:** "Kohler West and Horrall Glaciers, which flow at speeds of 715 ± 319 m/yr and 401 ± 173 m/yr at the grounding line in 2022, respectively."

**Edit Line 133:** "which flow at speeds of 772 ± 211 m/yr, 203± 93 m/yr and 810 ± 169 m/yr, respectively (Table 1)."

**Edit Line 135:** "Kohler East and Smith West Glacier where ice speeds increased by 31.7 m/yr$^2$ and 31.5 m/yr$^2$ from 2005 to 2022, respectively (Table 1)."

| 53 | Line 128: To focus purely on the glaciers, I suggest removing the clause "to the east of Crosson ice shelf." | Done.

**Edit Line 134:** "Ice is discharged through Pope, Vane, and Haynes Glaciers" |
| --- | --- | --- |
| 54 | Figure 2 caption – see suggestions for figure 1 caption. | Done. We have amended the caption in line with the previous comment.

**Edit Figure Caption 2:** "a) Sentinel-1 SAR derived average speed 2015-2022. The 2011 grounding line location (solid black line) (Rignot et al., 2016) and the location of the 8 flow line profiles (dashed black lines) are also shown." |
| 55 | Table 1 – I suggest adding which ice shelf the glacier feeds in the "region" column. I suggest bolding the glacier that slowed down (lines 136-137). | Done. We have added the suggested column.

**Edit Table 1:** |

| Region | Ice Stream Name | Ice Shelf the Ice Stream Primarily Feeds | Mean Ice Speed in 2022 (m/yr) | Rate of speed change 2005-2022 (m/yr²) | Rate of speed change for Sentinel-1 period (2015-2022) (m/yr²) | Speed change from 2005-2022 (%) | Speed change in Sentinel-1 period (2015-2022) (%) | Rate of Surface Elevation Change (m/yr) |
| --- | --- | --- | --- | --- | --- | --- | --- | --- |
| Western | Horrall | Dotson | 401 ± 173 | 2 | 4 | 10 | 9 | -0.8 |
| | Kohler West | Dotson | 715 ± 319 | -5 | -2 | -10 | -3 | -1.1 |
| Central | Kohler East | Dotson | 1215 ± 275 | 32 | 7 | 84 | 5 | -2.5 |
| | Smith West | Crosson | 1188 ± 253 | 32 | 11 | 87 | 8 | -2.6 |
| | Smith East | Crosson | 1093 ± 416 | 14 | 0 | 30 | 0 | -2.7 |
| Eastern | Pope | Crosson | 772 ± 210 | 3 | 1 | 7 | 1 | -2.2 |
| | Vane | Crosson | 203 ± 93 | 5 | 5 | 76 | 22 | -0.9 |
| | Haynes | Crosson | 810 ± 169 | 17 | 10 | 60 | 11 | -1.1 |

| 56 | Section 3.2: It is not clear why comparing 2005-2022 and 2015-2022?

Why is the percentage speed change the best way to report these results?

I also suggest adding a sentence about how you calculated the change in speed in the methods or supplemental information. | Done.

We felt that the percentage speed change is more indicative of the magnitude of change each ice stream is undergoing. For instance, Vane Glacier flows at ~200 m/yr in 2022 with a rate of change of 5 m/yr$^2$, which may sound small compared to the faster flowing |

| | | |
|---|---|---|
| | Also, is the rate of speed change from 2005-2022 including the Sentinel 1 velocities or only the measures product? | ice streams of 32m/yr (Kohler East and Smith West). However, when considering the percentage change, it equates to 76% for Vane over the study period and therefore the magnitude of change is not too much smaller than Kohler East and Smith West 84-87%.

We have added a sentence describing the speed change method in the main paper.

**Edit Line 138:** "We used our ice velocity measurements to fit a linear trend in each pixel to calculate the rate of change in ice speed from all speed data 2005 to 2022 and for the Sentinel-1 period 2015-2022 across the PSK region." |
| 57 | Line 151: Can you specify what you mean by increased 1.5 times from 2015 to 2022? Also, there is a not a 2005-2015 mean in Table 1. | Done.

**Edit L155:** "On Kohler West Glacier, the rate of ice flow slow-down has increased since 2015 (- 8 m/yr$^2$) compared to the 2005 to 2015 mean (- 4 m/yr$^2$) (Table 1)." |
| 58 | Line 152: I suggest using a different word than "around" in this context. | Done.

**Edit Line 159:** "short-term speed variability in addition to the 17.5-year trend" |
| 59 | Line 155: How are you defining "short term"? Depending on the audience, some might say "from 2005 to 2010 and from 2014 to 2017" are longer term changes. | Done.

**Edit Line 159:** "short term (sub decadal)" |
| 60 | Figure 3: Overall, this is a nice figure. Can you detail in the methods section how you detrended the velocity?

I suggest including in the caption what the yellow shading is. It is difficult to tell which glaciers are which color. Please use different distinct colors for each glacier instead.

Previously, you say it is a 17.5 year study period so I am confused here.

I also wonder if it I better to use the average of multiple years (say 2005-2008) as a reference | Done.

**Edit L104:** "To explore the shorter-term (sub-decadal) variability data were detrended by subtracting the linear trend."

**Edit Figure 3 Caption:** The shaded regions represent the rapid periods of speed up demonstrating the periodicity around the 17.5-year trend.

**Edit Figure 3:** |

| | | |
|---|---|---|
| | year due to the possibility that 2005 could be anomalous in any way. |  |
| 61 | Line 164: Which panel in Figure 4? | Done.

**Edit Line 171:** "2019 (Fig. 4d)" |
| 62 | Figure 4: The aesthetic of this figure is very nice! Try to make the other figures more like this one! Please do make the panel letters larger and more in the corner. | Done.

**Edit Figure 4:**
 |
| 63 | Lines 163-165: Is it possible to determine whether the thinning occurred first or that the thinning was a result of the ice flow change?

Can you comment on the possibility that the mass input changed, or the ice shelf thinning cause a decrease in buttressing stress?

It also looks like the thinning is widespread in both the Smith (East and West) and Kohler (East and West) ice streams, according to Figure 1d. | The PSK region has been observed to be thinning since at least the 1970s (Shepherd et al., 2019a). It is likely these ice dynamic changes initiated in or before the 1940's, particularly on Pine Island Glacier r (Clark et al., 2024; Davies et al., 2017; Rignot et al., 2014; Shepherd et al., 2019b; Smith et al., 2016). Ice speed responds near instantaneously to changes in driving factors, whereas thinning changes takes longer to propagate through. It is challenging to disentangle the pattern of changes due to the highly complex potential driving factors and future detailed investigation would be of great benefit.

Please also see response to comment 21 where we amended the text to highlight the geometry and thinning gradient unique more clearly to Kohler West and Kohler East are the driving the piracy. |

| | | |
|---|---|---|
| 64 | Line 172: There is a new paper by G. Collao-Barrios that discusses a different kind of flow variation that is tidally driven. I suggest including it somewhere in this section. Also, I suggest starting a new paragraph here. | We thank the reviewer for their suggestions and feel the edits in response to comment 18 have resolved the need for a new paragraph here.

The Callao-Barrios paper is of great interest, however, is currently only a preprint which was published after the submission of this paper, so we have not cited it here. |
| 65 | Line 173: Can you elaborate on this sentence and where "elsewhere" is? | Done.

**Edit Line 252**: "These changes in flow direction are consistent with the observed changes in surface elevation and ice thickness in this region and would be expected to occur elsewhere in places where the geometry permits, such as at Totten and Vanderford Glaciers (McCormack et al., 2023)." |
| 66 | Line 174-177: I suggest including this in the discussion and clarifying what you mean. This has not occurred on ice caps/glacier scale in a15 year period or it HAS? It is unclear in the current wording. I also suggest incorporating these references in this portion of the text: Catania et al., 2012 - https://doi.org/10.3189/2012JoG11J219 and Conway et al., 2002 https://doi.org/10.1038/nature01081 | Done. We have clarified the wording and added the two suggested references.

**Edit L258**: "To our knowledge, however, redirection of ice flow from one ice stream to another in Antarctica has not been observed directly on short ~15-year timescales, but it may have occurred at glacier and ice cap scale."

**Edit L256:** "reconstructions (Catania et al., 2012; Conway et al., 2002; Conway and Rasmussen, 2009; Iizuka et al., 2010)" |
| 67 | Line 178: Please take a look at these references and revise this sentence: Conway et al. 2002; and Price et al, 2001 https://doi.org/10.3189/172756501781832232 | Done. We thank the reviewer for highlighting these papers to us and have edited the text to clearly distinguish that we are highlighting the these are the first direct observations of such a rapid timescale of change not that the phenomenon has not occurred before.

**Edit L258: "**To our knowledge, however, redirection of ice flow from one ice stream to another in Antarctica has not been observed directly on short |

| | | ~15-year timescales, but it may have occurred at glacier and ice cap scale." |
|---|---|---|
| 68 | Figure 4: This is an interesting plot (4b)! However, the axis label should read 'downstream flux', correct? Smith East seems a bit suspect - is that far upstream increase in flux due the small speed increase? I guess I'm saying, check the numbers for this one specifically, the others look good!

Also, are there 'no data' areas in here? What do the pure white patches mean? | Done. We thank the reviewer for highlighting this disparity. We have added a short description to the methods section on how we calculated the upstream flux and to clarify this is what the figure is showing. Please see the response to comment 12.

The pure white patches are 'no data' areas. |
| 69 | Line 190: I'm not sure there is a clear compressive arch for Crosson, or if so, how are you defining it? Do you have strain rate data? I think it's best to just say "inland." | Done. The 1996 compressive arch was defined in Lilien et al., 2018, we have edited to the text to more clearly reflect this.

**Edit L188:** "inland of the 1996 compressive arch (Lilien et al., 2018) at the eastern shelf edge and becoming heavily crevassed." |
| 70 | Lines 191-194: This rewording is confusing, consider revising. Additionally, why is the "distance decreasing from the island" of importance? | Done.

**Edit L189:** "During the same period as these large calving events, a 5 km rift formed at the eastern margin of the Crosson Ice Shelf and grew to ~12 km across the width of the ice shelf towards Bear Island during the study period (Fig. 5b, S1and S3). The distance between the end of the rift and Bear Island decreased from 8 km in 2015 to 5 km in 2022, with the rate of rift growth slowing from 2016 to 2022. If the rift eventually reaches Bear Island a ~4 km$^2$ iceberg may be released." |
| 71 | Line 194: How did you measure the increase in damage? | Done.

**Edit L198:** "The substantial visible increase in damage on the Crosson Ice Shelf and calving front extending inland of the 1996 compressive arch will have reduced the buttressing strength provided by the ice shelf, contributing to the observed speed-up of the glaciers that drain into it (Fig. 5)." |
| 72 | Figure 5: See general comments. | Done. |

| | | **Edit Figure 5:** |
|---|---|---|
| | |  |
| 73 | Line 220: This slow down is difficult to see in Figure 3. | Done. We have amended the text to more clearly indicate it's a slower rate of speedup rather than slowdown.

**Edit L222:** "We note periods of rapid speed-up 2005 to 2011 with an average speed difference across all ice streams of 14 % and during 2014-2017 (12 %) were interrupted by periods of reduced rates of speed up between 2011-2013 (4 %) and 2017-2020 (2 %) (Fig. 3a); the timing and relative magnitude of these speed fluctuations vary between ice streams, but the absolute magnitude of the speed change is similar across the region (Fig. 1). |
| 74 | Lines 225-231: This section is very lengthy and needs to be more concise. Can you be more specific what the hydrographic surveys and numerical models show?
• It is necessary to include the location of PIG on one of your figures so the readers know where the glacier is relative to PSK.
• PIG also has a very unique geometry compared to PSK (in a relatively narrow embayment compared to Thwaites/DotsonCrosson/etc); is it reasonable to assume the same oceanographic mechanisms are happening there?
• From 2017 onwards, Kohler East/Smith West Vane/Haynes (Can't tell the color differences) continued to speed up.
• What would cause this difference between all of these glaciers, is the acceleration different? | Done. We have addressed the reviewer comments as best we can. It can be difficult to get the right balance between providing a good level of detail in the discussion and covering all the necessary points. We hope we have reached a good middle ground here.

**Edit Figure 1**: PIG and ASE labels added.

We agree with the reviewer that PIG & PSK are not the same, however as they are both located in West Antarctica, and they are impacted by the same ocean we felt it was useful to discuss the two regions in relation to each other.

**Edit L236:** "Particularly, the rate of speedup decreasing after around 2020 may also be linked to the advance in the floating ice in front of Haynes Glacier which partially recloses the pathway of open water accessing the eastern side of Crosson, perhaps limiting the penetration of the warmer water into the cavity. A re-buttressing |

| | | of Crosson may have also occurred, albeit in a new configuration, with the calving front connecting to Mount Murphy acting as a stabilising pinning point and lowering the rate of observed speed up." |
|---|---|---|
| 75 | Line 238: how does the 82 m/yr compare to your observed data | Done.

**Edit L248:** Indeed, 18 m of thinning and a 0.03-degree decrease in ice surface slope (comparable to those observed on Kohler West) would cause an 82 m/yr decrease in ice velocity, which is similar to the observed slow-down from 2005 to 2022 (88 m/yr) (Equation S1, Cuffey and Paterson, 2010). |
| 76 | Line 240-244: see previous comment | Comment. Apologies, we are not completely clear what the reviewer is requesting in this comment. We can address the point if further detail is provided? |
| 77 | Figure S2: Can you include your method on how you calculated "damage" and how you defined it? In the main text as well as the supplemental information | Please see response to comment 17. |
| 78 | Line 246-247: Please include details on how calculated the mass flux? What did you use for bedrock/ice thickness? | Please see response to comment 12. |
| 79 | Line 245-250: I suggest can marking (maybe on fig. 4) where the ice shelf divide migrated from and by how much it migrated by? | Comment. We show that there is a change in ice flow direction, and the flow vectors plotted on figure 4 show that this would modify the location of the ice divide. We don't delineate a new drainage basin for the PSK glaciers in this paper, so we have not drawn on new drainage basin boundaries for that reason. |
| 80 | Line 256-258 – The paper you cite suggest that grounding line retreat will resume. "Ice speed did not decrease until 2012 when the cold-water anomaly began. The slowdown was likely also partially a result of the advection of thicker ice onto the bathymetric ridge [Joughin et al., 2016]; the lower ocean heat content likely lowered basal-melt rate, allowing thicker ice to advect farther downstream. As the water in the cavity subsequently warmed, however, speeds increased to their precold anomaly rates, suggesting grounding line retreat will resume." I think the use of "thermocline" is misleading, as it was cold water that entered the cavity below | Done. We thank the reviewer for their useful discussion. We have edited the text to reflect their comments.

**Edit L280:** "At Pine Island Glacier, the rapid reduction in ice velocity in response to the cold-water intrusion and presumably reduced basal melt rates has been used as evidence to suggest that MISI is not underway, or at least is conditional upon continued ocean forcing (Christianson et al., 2016)." |

| | | |
|---|---|---|
| | PIG, slowing it less than <4%, then when the warmer water came back the speed went back to accelerating. Though recent modelling studies suggest PIG's GL may be temporarily stable, it's continued thinning and acceleration suggest otherwise. I would not say the Christianson paper agrees with the Hill 2023 paper. They have fundamentally different conclusions.

Additionally, to say that none of the glaciers in the PSK region are susceptible to MISI on shorter timescales disregards the recent results by Reed et al., 2024 on the significant GL retreat from 1970-1990 (https://www.nature.com/articles/s41558-023-01887-y) | **Edit L282:** "Recent modelling efforts suggest that the current grounding line positions in the ASE are stable (Hill et al., 2023)."

Reed et al. is highly relevant to our study, however, as the paper was only just published after the completion of our manuscript, the results of this paper ultimately didn't inform our study. We look forward to discussing the complimentary nature of both papers with Brad and his co-authors at the next conference we all attend, and I have no doubt that his paper will be widely cited. |
| 81 | Line 261: I suggest rephrasing this sentence as the wording is confusing. | Done.

**Edit Line 285**: However, we observe that ice streams on prograde (Kohler West) and retrograde bed topography are responding with opposing speed change trends (Fig. 2). |
| 82 | Lines 255-268: This part of the discussion needs to be revised to be organized and include clear key takeaways. Please see the suggestions in the general comments. | Please see response to comment 19. |
| 83 | Morlighem 2017 is not in the reference list. | Done.

**Edit Line 490:** Morlighem, M., Williams, C. N., Rignot, E., An, L., Arndt, J. E., Bamber, J. L., Catania, G., Chauché, N., Dowdeswell, J. A., Dorschel, B., Fenty, I., Hogan, K., Howat, I., Hubbard, A., Jakobsson, M., Jordan, T. M., Kjeldsen, K. K., Millan, R., Mayer, L., Mouginot, J., Noël, B. P. Y., O'Cofaigh, C., Palmer, S., Rysgaard, S., Seroussi, H., Siegert, M. J., Slabon, P., Straneo, F., van den Broeke, M. R., Weinrebe, W., Wood, M., and Zinglersen, K. B.: BedMachine v3: Complete Bed Topography and Ocean Bathymetry Mapping of Greenland From Multibeam Echo Sounding Combined With Mass Conservation, Geophys Res Lett, 44, 11,051-11,061, https://doi.org/10.1002/2017GL074954, 2017. |

| 84 | Equation S1 – please format the units correctly, like you did for the others: | Done.

**Edit:** A = rate factor for ice ($9.3e-25\ s^{-1}Pa^{-3}$) |

---

## Referee Report (RR1)

**Review of "Speed-up, slowdown, and redirection of ice flow on neighbouring ice streams in the Pope, Smith and Kohler region of West Antarctica"**

I thank Selley et al., for taking into consideration all of my (and other reviewers) suggestions. The authors made a great effort at revising the manuscript to incorporate all of the comments. There are some remaining comments to be addressed, which, when done so, I am sure the manuscript will be ready for publication.

My only major comments are on the discussion sections 4.3 and 4.4. I think section 4.3 would be improved upon if the authors incorporated more studies to strengthen their speculations regarding the stability of Dotson and Crosson ice shelves and/or included other potential mechanisms that could be at play (though not analyzed in this study). For section 4.4, I still found the discussion around MISI to be unclear. Could it happen in the PSK region? If so, on what timescale (Reed et al., 2024 suggests decadal timescale is possible)? Please also see my comment in the section below.

Minor comments:

Lines 46-48 – is the time period of the basal melt rate the same as the thinning rate? If not, can you specify? Thanks!

115-116 – Here, you say you have annual resolution for the 17.5 year study period but in the sentence before (line 112) you say 2009-2019, so what did you use for the other 7.5 years?

128-130 – These lines may fit better in the methods section.

138 – I would find it helpful if you have "Measures period 2005-2022" in this section, since the other dataset is referred to as "Sentinel-1period 2015-2022" for consistency and clarification.

198-199 – I would suggest including more of this discussion regarding damage and buttressing in your discussion section. (See comment for lines 238-240).

Figure 5 – The figure caption is missing information on panel C. Also, for panel C the color of the font, location of the rift outlines are a bit confusing, I suggest moving the text of the years and/or making the text a different color.

222-225 – Here, the font size changed in text.

238-240 – Here you discuss the potential of Crosson, rebuttressing, however in the results (see comment on lines 198-199) you mention that there is potentially reduced buttressing from damage. How do you reconcile these two potential results and can you expand a bit more in this paragraph on that?

Discussion is overall much easier to read with new subheadings and strengthened text.

Section 4.2 – I really appreciate how you've strengthened this paragraph!

Section 4.3 – I agree that your results are compelling, regarding the (de)/stabilization of the ice shelves, however, I would like to see a bit more discussion about other potential mechanisms (e.g. basal melting, wave action, etc.) and how the migration of the ice divide interplays with those other processes. Though this paper below is still a preprint, I don't suggest citing it, but rather using it to inform more of your discussion on this topic, perhaps they have relevant references for this section of your discussion.

Wild, C., et al. "A Tale of Two Ice Shelves: Competing Glacial Dynamics During the Unpinning of the Dotson-Crosson Ice Shelf System, West Antarctica." *Authorea Preprints* (2024). DOI: 10.22541/essoar.172745052.28786721/v1

Lines 286-288 – As stated in my last review, I really think it's important for the authors to address the decadal timescale of which MISI occurred on PIG that Reed et al., 2024 investigates. I am only requesting a few sentences and to tone-down the conclusion that MISI may not be a major dynamical factor at short timescales, especially when this is not a primary component of the study. I am disappointed to see that they did not address this in the last round of revisions. I strongly encourage them to do this when they next revise the paper.

Line 301 – I would suggest adding the line about thinning reducing driving stress earlier in the discussion and not only including it in the conclusion.

---

## Author Response (AR2)

**Response to Reviewer Comments - Speed-up, slowdown, and redirection of ice flow on neighbouring ice streams in the Pope, Smith and Kohler region of West Antarctica**

We thank the reviewers for their time and effort in reviewing our paper, "Speed-up, slowdown, and redirection of ice flow on neighbouring ice streams in the Pope, Smith and Kohler region of West Antarctica", submitted for publication in the cryosphere. We welcome the positive feedback and insightful comments which we have endeavoured to fully address in this resubmitted revision, and we hope you agree this improves the manuscript. We have incorporated the suggestions made. The changes are highlighted in the manuscript. Please see below a point-by-point response to the reviewers' comments, where all line numbers refer to the revised manuscript file.

| Line | Comment | Response |
|------|---------|----------|
| **Reviewer #1** | | |
| 1 | My comment #7 around agreement with MEaSUREs. Thanks for the explanation here. Do you know why the differences are so large for the Vane Glacier, and is it significant? Did the authors include in the updated manuscript a comment on the data in the table provided in their response document? I couldn't see it, but I think both the table and the text provided would be helpful additions to the supp info. | **Done.** We thank the reviewer for their suggestion of including the additional information in the supplementary materials. We have added this new table.

Regarding Vane Glacier, it is a much smaller ice stream (~half the size and speed of the next smallest stream) and is highly constrained by the surrounding topography. As described in the paper the errors in the ice speed products are higher in highly crevassed regions such as shear margins and can be a characteristic of smaller outlet glaciers such as Vane where the tracking algorithms don't perform as well. We see this in our spatially variable error estimate; however, it is not captured in the MeASUREs error product. The spatial resolution of both ice speed datasets is different, so given the smaller width of this ice stream there may be an element of spatial aliasing where slower moving areas are incorporated into the mean around the central faster flowing stream. This is more likely to affect the coarser resolution MeASUREs product so we would expect the difference to be greater here.

We used a linear fit in all ice speed trend analyses to minimise the impact of any offset between the two speed products.

**Edit Supplementary:** We have now included the response to comment 7 in the supplementary information including adding the comparison table as Table S3. |

**Comparison of the Sentinel-1 and MeASUREs ice speed data overlap period between 2016 and 2017**

We compared the two ice speed datasets in this time period in the 2.5 km diameter regions at the grounding line of all 8 glaciers (Table S3). Overall, the Sentinel-1 velocity measurements are slightly faster than MeASUREs result on all glaciers, with an average speed difference of 21 m/yr (5%) in 2016 and 17 m/yr (3 %) in 2017. If we remove the slower flowing Horrall and Vane glaciers which flow at 401 and 203 m/yr respectively, the absolute difference reduces slightly (19 and 13 m/yr respectively), but the percentage difference reduces substantially to 2 and 1 % for 2016 and 2017 respectively. This is well within the error on our speed measurements.

The majority of this speed difference is likely caused by differences in the underlying spatial resolution of the satellite data and the step and window size used for the feature tracking. It is well known (Lemos et al., 2018) that finer spatial resolution satellite datasets allow you to track ice speeds at high spatial resolution which then detect small regions of fast flow. Equally using a larger window and step size in the feature tracking step will tend to effectively smooth the output ice speed result which subtly reduces the average mean speed, and it also tends to increase spatial coverage slightly. As we move into an era with more SAR satellites that enable us to track ice speed at different resolution from different sensors for any one location, it will be increasingly important to characterise these differences, in the same way the satellite altimetry community is already doing for laser and multi-frequency radar altimetry products.

There is also a slight difference in the time periods covered by the two products: Sentinel runs Jan-Dec whereas MeASUREs runs from July to June. We know from IMBIE studies that any difference in the spatial and temporal domain of different datasets can result in differences between them. These differences may be due to error in the products; however, they may also be due to real geophysical change that

occurs between the time periods. We used a linear fit in all ice speed trend analyses to minimise the impact of any offset between the two speed products.

**Table S3**. Comparison of the Sentinel-1 and MeASURES ice speed data overlap period between 2016 and 2017

| Time Period | Horrall | Kohler West | Kohler East | Smith West | Smith East | Pope | Vane | Haynes | Average (all) | Average (fast flowing) |
|---|---|---|---|---|---|---|---|---|---|---|
| MeASUREs minus S1 (m/yr) | | | | | | | | | | |
| 2016 | -17 | -18 | -28 | -8 | -28 | -14 | -42 | -15 | -21 | -19 |
| 2017 | -33 | -3 | -28 | -12 | -6 | -2 | -21 | -29 | -17 | -13 |
| Average S1 Speed (m/yr) | | | | | | | | | | |
| 2022 | 401 | 715 | 1215 | 1188 | 1093 | 772 | 203 | 810 | 800 | 966 |
| Difference between Measures and S1 speeds as % of 2022 speed | | | | | | | | | | |
| 2016 | -4 | -3 | -2 | -1 | -3 | -2 | -20 | -2 | -5 | -2 |
| 2017 | -8 | 0 | -2 | -1 | -1 | 0 | -10 | -4 | -3 | -1 |

| 2 | My comment #14. Sorry, I'm still a bit hazy on the edit at L220. Do you mean something like: "We note periods of rapid speed-up 2005-2011, with an average speed difference across all ice streams of 14%. During 2014-2017, the average speed increased by 12%, although there were periods of slow down between 2011-2013 (4%) and 2017-2020 (2%)…" Perhaps the speed up percentages could be reported as +% and the slow down negative -% to better differentiate. | **Done.**

**Edit L162:** 2011 to 2013 (-4%) and from 2017 to the end of the study period in 2022 (-2%) |
|---|---|---|

**Reviewer #1**

| 3 | I thank Selley et al., for taking into consideration all of my (and other reviewers) suggestions. The authors made a great effort at revising the manuscript to incorporate all of the comments. There are some remaining comments to be addressed, which, when done so, I am sure the manuscript will be ready for publication. My only major comments are on the discussion sections 4.3 and 4.4. I think section 4.3 would be improved upon if the authors incorporated more studies to strengthen their speculations regarding the stability of Dotson and Crosson ice shelves and/or included other potential mechanisms that could be at play (though not analyzed in this study). For section 4.4, I still found the discussion around MISI to be unclear. Could it | **Comment.** We thank the reviewer for their insight and suggestions on the manuscript. We have endeavoured to strengthen the sections they're highlighted and have and incorporated their suggestions as outlined in the responses below. |
|---|---|---|

| | | |
|---|---|---|
| | happen in the PSK region? If so, on what timescale (Reed et al., 2024 suggests decadal timescale is possible)? Please also see my comment in the section below. | |
| 4 | Lines 46-48 – is the time period of the basal melt rate the same as the thinning rate? If not, can you specify? Thanks! | **Done.**

**Edit Line 48:** "with average basal melt rates of 5.4 ± 1.6 and 7.8 ± 1.8 m/yr between 1994 and 2018 (Adusumilli et al., 2020)," |
| 5 | 115-116 –Here, you say you have annual resolution for the 17.5-year study period but in the sentence before (line 112) you say 2009- 2019, so what did you use for the other 7.5 years? | **Done.**

**Edit Line 115:** "2005 to 2022 using Moderate Resolution Imaging Spectroradiometer (MODIS) imagery" |
| 6 | 128-130 –These lines may fit better in the methods section. | **Done.**

**Edit Line 105:** "For time series analysis, we extract ice speed measurements along flowline transects located on the fast-flowing central trunk of all 8 ice streams (Fig. 1a and Fig. 2) and compute grounding zone time-series averaging within 2.5 km diameter circles where the flow lines intersect with the grounding line (Rignot et al., 2016)." |
| 7 | 138 – I would find it helpful if you have "Measures period 2005-2022" in this section, since the other dataset is referred to as "Sentinel-1 period 2015-2022" for consistency and clarification. | **Done.**

**Edit Line 139:** "from all speed data (MeASUREs and Sentinel-1) 2005 to 2022 and for the Sentinel-1 period 2015-2022 across the PSK region" |
| 8 | 198-199 – I would suggest including more of this discussion regarding damage and buttressing in your discussion section. (See comment for lines 238-240). | Please see response to comment 11. |
| 9 | Figure 5 –The figure caption is missing information on panel C. Also, for panel C the color of the font, location of the rift outlines are a bit confusing, I suggest moving the text of the years and/or making the text a different color. | **Done.** We thank the reviewer for highlighting this oversight and have amended the caption. Regarding the recommendations for the rift colours/text we have tried to make our figures as clear as possible and the text describes the narrative of the crack formation and therefore, we have not changed the figure further. We can assure the reviewer that we have tested many options here and we do feel the current version displays the information most clearly.

**Edit Figure Caption 5:** "**(B)** Zoom of Crosson Ice Shelf calving front locations from 2005-2022. **(C)** Location of the rift approaching bear island through the Sentinel-1 period (2015-2022). |

| | | |
|---|---|---|
| 10 | 222-225– Here, the font size changed in text | **Done.**

**Edit Line 222-225:** Font size changed to match. |
| 11 | 238-240 –Here you discuss the potential of Crosson, rebuttressing, however in the results (see comment on lines 198-199) you mention that there is potentially reduced buttressing from damage. How do you reconcile these two potential results, and can you expand a bit more in this paragraph on that? | **Done.** We have provided additional discussion of the potential variation in buttressing due to damage but also potential pinning points as the reviewer recommended.

**Edit L241:** Change in ice shelf pinning can cause destabilisation through reducing the structural stability of the shelf when they are in advanced stages of thinning (Benn et al., 2022) but also by potentially initiating calving events (Arndt et al., 2018). These processes may impact the ice shelf at different times in different regions, potentially driving some of the observed speed change. |
| 12 | Discussion is overall much easier to read with new subheadings and strengthened text. Section 4.2 –I really appreciate how you've strengthened this paragraph! | **Comment.** We thank the author for their insights and suggestions which led to the strengthened discussion. |
| 13 | Section 4.3 – I agree that your results are compelling, regarding the (de)/stabilization of the ice shelves, however, I would like to see a bit more discussion about other potential mechanisms (e.g. basal melting, wave action, etc.) and how the migration of the ice divide interplays with those other processes. Though this paper below is still a preprint, I don't suggest citing it, but rather using it to inform more of your discussion on this topic, perhaps they have relevant references for this section of your discussion:
Wild, C., et al. "A Tale of Two Ice Shelves: Competing Glacial Dynamics During the Unpinning of the Dotson-Crosson Ice Shelf System, West Antarctica." Authorea preprint (2024) | **Done.** We thank the reviewer for highlighting this preprint to us and offers further insight to the dynamics at Crosson and Dotson. However, as this preprint only became available several months after the submission of our paper it did not inform the results of this study which is why we haven't cited it. We look forward to seeing the final published version of his paper and to having discussions about the complementary nature of our work.

We have taken this feedback onboard and added some text on further potential mechanisms.

**Edit Line 280: "**There has been significant variability in sub-shelf melt rates in the Crosson Ice Shelf spatially and temporally over the past 20 years, with melt peaking in the early 2010s (Jenkins et al., 2018). Additionally changes to pinning points can further cause destabilisation through initiating calving events (Arndt et al., 2018) and in advanced stages of thinning (Benn et al., 2022). Further work exploring the chronology of changes to stress structures on the Crosson Ice Shelf would be invaluable to further establish the interplay of these driving mechanisms." |

| 14 | Lines 286-288 –As stated in my last review, I really think it's important for the authors to address the decadal timescale of which MISI occurred on PIG that Reed et al., 2024 investigates. I am only requesting a few sentences and to tone-down the conclusion that MISI may not be a major dynamical factor at short timescales, especially when this is not a primary component of the study. I am disappointed to see that they did not address this in the last round of revisions. I strongly encourage them to do this when they next revise the paper. | Done. We thank the reviewer for their insight and have amended the text to include Reed et al., 2024 and nuance the strength of the MISI argument. We do think there is value in including MISI discussions as it has the potential to impact the glaciers in this region and we think our results will be a useful contribution to future studies on this topic.

**Edit Line 297: "**short (sub-decadal) timescales across the PSK region.  Conversely, more recent modelling work based at Pine Island Glacier indicates irreversible grounding line retreat between the 1940s and the 1990s is an example of MISI with recent changes being primarily driven internally (Reed et al., 2024)."

**Edit Line 305**: "or it could be driven by internal ice dynamic feedbacks (such as MISI)(Reed et al., 2024)" |
| --- | --- | --- |
| 15 | Line 301 – I would suggest adding the line about thinning reducing driving stress earlier in the discussion and not only including it in the conclusion. | **Done.**

**Edit line 252: "**The region continues to thin and therefore thinning induced reductions in driving stress may have contributed to the observed slow down." |

---

## Author Response (AR3)

**Response to Editors Comments - Speed-up, slowdown, and redirection of ice flow on neighbouring ice streams in the Pope, Smith and Kohler region of West Antarctica**

**Response to Editor:**

We thank the editor for their time and effort in reviewing our paper, "Speed-up, slowdown, and redirection of ice flow on neighbouring ice streams in the Pope, Smith and Kohler region of West Antarctica", submitted for publication in the cryosphere. We welcome the positive feedback and insightful comments which we have endeavoured to fully address in this resubmitted revision, and we hope you agree this improves the manuscript. We have incorporated the majority of the suggestions made, and in the limited cases where we have not, we have provided a detailed description of the justification foreach decision. The changes are highlighted in the manuscript. Please see below a point-by-point response to the editors comments, where all line numbers refer to the revised manuscript file.

| Line | Comment | Response |
|---|---|---|
| 19 | Add ice | **Edit Line 19:** "ice sheet" |
| Fig 1 | Add Crosson and Dotson Labels to ice shelves | **Edit Figure 1:**

**Edit Fig 1 Caption:** Crosson Ice Shelf (CIS) and Dotson Ice Shelf (DIS) |
| Fig 1 Caption | Remove space 2015 - 2022 | **Edit Figure 1 Caption:** "2015-2022" |
| Fig 1 Caption | Correct the panel lettering | **Edit Figure 1 Caption:** "(B) Rate of elevation change for 1992 to 2023 (red) (Shepherd et al., 2019) and grounding line locations from 1992 to 2020 (blue) (Milillo et al., 2022; Rignot et al., 2016). (C) Observed rate of speed change over the full 17.5-year study period, from 2005/05/01 to 2022/05/30. (D) Observed rate of speed change during the Sentinel-1 period from 2015/06/01 to 2022/05/31. Measurements are superimposed on BedMachine bedrock topography (Morlighem et al., 2017)." |
| 92 | Is threshold the correct word here? Factor? | To maintain consistency with other published work using the same method we have kept the threshold terminology. |
| 139 | Repetition | **Edit Line 138:** We used our ice velocity measurements to fit a linear trend in each pixel to calculate the rate of change in ice speed from all speed data (MeASUREs and Sentinel-1) 2005 to 2022 and for the Sentinel-1 period 2015-2022 across the PSK region (Fig. 1b and 1c). |
| 186 | Fig. 4 should be Fig. 5 please check all numbering | **Edit L186:** "(Fig. 5)." |
| Fig 5 Caption | You use capital letters here please check all figure captions for consistency | **Edit Figure Caption 2:** "Figure 2: (A) Sentinel-1 SAR derived average 2015 to 2022 ice speed over the PSK region. The 2011 grounding line location (solid black line) (Rignot et al., 2016) , the location of the 8 flow line profiles (dashed black lines) and |

| | | the intersection 2.5 km buffer (purple circles) are also shown. **(B)** Observed ice speed from 2005 to 2022 at the 2011 grounding line (Rignot et al., 2016) of Dotson and Crosson Ice Shelves in the Amundsen Sea Sector of West Antarctica. The x-axis is shown as distance from the grounding line, with positive values indicating the inland section of the profile on the ice sheet, and negative values indicating seaward locations. Also shown are the intersections of flow lines (black vertical lines), the bed topography (brown shading) and ice surface elevation and thickness (grey shading) from BedMachine (Morlighem et al., 2017) . Observed speed along the flowlines **(C)** Horrall **(D)** Kohler West, **(E)** Kohler East, **(F)** Smith West, **(G)** Smith East, **(H)** Pope, **(I)** Vane and **(J)** Haynes." |
|---|---|---|
| 208 | Focused? | **Edit Line 208:** "focused" |
| 222 | It's difficult to reconcile these percentages with the 51% reported in the conclusion. I assume the numbers are, ok? A few words to make sure readers are not puzzled and can reconcile them? | We have included the signal sign to match the results description which hopefully makes it clearer. **Edit line 223:** "We note periods of rapid speed-up 2005 to 2011 with an average speed difference across all ice streams of 14 % and during 2014-2017 (12 %) were interrupted by periods of slow down between 2011-2013 (-4 %) and 2017-2020 (-2 %) (Fig. 3a); the timing and relative magnitude of these speed fluctuations vary between ice streams, but the absolute magnitude of the speed change is similar across the region (Fig. 1)." |
| 241 | Maybe Miles and Bingham (2024) is relevant to cite here or just below? | **Edit Line 244:** (Miles and Bingham, 2024) |
| 264 | I did not find the two parts of this sentence easy to articulate. Not sure what the second member really means | **Edit Line 264:** "To our knowledge the redirection of ice flow from one ice stream to another has not been observed directly on ~15-year timescales, although, it may have occurred at glacier and ice cap scales." |
| 285 | Delete " | **Edit Line 285:** Removed. |
| 292 | Elsewhere in the article you focus on the acceleration of PIG (when discussing "damage"). Maybe provide the time period of speed reduction to avoid confusion | **Edit Line 292:** "At Pine Island Glacier, the rapid reduction in ice velocity in response to the cold-water intrusion in 2012 to 2013" |
| 295 | The ASE acronym was defined indeed but a long time ago. Consider spelling out. | **Edit Line 295:** "Amundsen Sea Embayment (ASE)" |
| Supplementary | Fig S2: it was not clear from the legend why you have two series of velocity maps from 2015-2022 and then 2005-2016. I suggest splitting this figure in two figures with the relevant explanation of the source in the legends (be careful to change the ref to these figures in the main). | **Edit Fig S2 Caption:** First shown are the Sentinel-1 ice speed maps 2015-2022 followed by the MeASUREs dataset 2005-2016. |

| Supplementary | Also in the supplement you mention "We know from IMBIE studies that any difference in the spatial and temporal domain of different datasets can result in differences between them." Did IMBIE really look into details at the differences in velocity products? (I though not, but I may be wrong). If not, no need to mention IMBIE here. | **Edit:** Removed. |
|---|---|---|
| Equation S1. | Ref to a text book? Cuffey and Patterson? | **Edit**: (Cuffey and Patterson, 2010) |